# An early mechanical coupling of planktonic bacteria in dilute suspensions

Simon Sretenovic[1], Biljana Stojković[2], Iztok Dogsa[1], Rok Kostanjšek[1], Igor Poberaj[3,4] & David Stopar[1]

It is generally accepted that planktonic bacteria in dilute suspensions are not mechanically coupled and do not show correlated motion. The mechanical coupling of cells is a trait that develops upon transition into a biofilm, a microbial community of self-aggregated bacterial cells. Here we employ optical tweezers to show that bacteria in dilute suspensions are mechanically coupled and show long-range correlated motion. The strength of the coupling increases with the growth of liquid bacterial culture. The matrix responsible for the mechanical coupling is composed of cell debris and extracellular polymer material. The fragile network connecting cells behaves as viscoelastic liquid of entangled extracellular polymers. Our findings point to physical connections between bacteria in dilute bacterial suspensions that may provide a mechanistic framework for understanding of biofilm formation, osmotic flow of nutrients, diffusion of signal molecules in quorum sensing, or different efficacy of antibiotic treatments at low and high bacterial densities.

[1] Biotechnical Faculty, University of Ljubljana, Vecna pot 111, Ljubljana 1000, Slovenia. [2] Medical Faculty, Institute of Biophysics, University of Ljubljana, Vrazov trg 2, Ljubljana 1000, Slovenia. [3] Faculty of Mathematics and Physics, University of Ljubljana, Jadranska 19, Ljubljana 1000, Slovenia. [4] Aresis Ltd., Ulica Franca Mlakarja 1a, Ljubljana 1000, Slovenia. Correspondence and requests for materials should be addressed to D.S. (email: david.stopar@bf.uni-lj.si)

Planktonic organisms are by definition untethered to surfaces and to each other[1]. It is therefore expected that at low cell densities in shear flow planktonic cells move independently. At the other extreme, in a biofilm, bacterial cells adhere to each other and/or to surfaces[2, 3]. Physiological properties of cells in biofilms are to a large extent determined by mechanical properties of the environment which are viscoelastic in nature and are supposedly missing in dilute bacterial suspensions[4, 5]. The progression from non-coupled to mechanically coupled cell network marks the beginning of cell coordinated behavior[6]. However, the transition to the interconnected network remains elusive and out of the reach of classical microscopic and rheological approach, as the early-tethered bacterial structures are fragile and dynamic.

A mechanical coupling over large distances has been demonstrated in biofilms, but not in dilute bacterial cultures[7]. In mature biofilms cells are mechanically coupled by a network of extracellular polymeric substances (EPS) that protect, retain water, organic compounds, inorganic ions and extracellular enzymes, permit redox activity, and facilitate horizontal gene transfer[8, 9]. With the development of biofilm, the extracellular matrix is gradually fortified by different components of EPS (i.e., polysaccharides, proteins, eDNA) which enable a transition from viscoelastic liquid to viscoelastic solid network[4, 5]. We have previously shown that increasing the concentration of eDNA causes a phase transition of levan, the major component of Bacillus subtilis biofilms, which starts to aggregate forming clusters of a few microns in size[10]. In general, biofilm structures that form upon colloidal self-assembly of constituent components at high concentrations are well documented and can be visualized by a variety of microscopic techniques[11–16]. Recently, we have reported on the physical interactions that facilitate self-assembly of B. subtilis cells at high cell densities into a mechanically coupled network employing dynamic rheology, small-angle X-ray scattering, dynamic light scattering, microscopy, densitometry, and sound velocity measurements in model viscoelastic polymer mixtures[17].

In this study, we probe viscoelasticity of dilute bacterial suspensions to clarify the link between extracellular polymer production and mechanical coupling of bacteria in the planktonic state. In dilute bacterial suspensions, one has to assess small viscosities near the viscosity of water at very small shear rates. In addition, the viscoelastic moduli of dilute bacterial suspensions are very low and are typically outside of the sensitivity of a classical rotational rheometer[18]. Recently, López et al.[19] demonstrated that classical rheology can be done at low shear rates at high cell densities (>1.1 × 10$^9$ cells per ml). The physical connections between cells at lower cell densities have not been tested yet. Using optical tweezers, we show that it is possible to detect long-range coordinated motion in bacterial clusters as well as weak mechanical coupling of bacterial pairs in dilute bacterial suspensions. We further show that mechanically coupled network of extracellular material is formed much earlier than expected. The phenomenon may have wide implications for understanding bacterial behavior and physiology in dilute suspensions.

## Results

**Mechanical coupling in bacterial clusters**. The correlated motion of neighboring bacteria in dilute bacterial suspensions using a single particle active microrheology was determined in the early exponential phase. When a single optically trapped bacterium in dilute suspension was moved, the neighboring bacteria followed the motion. Different bacterial dilute culture suspensions showed viscoelastic properties: *Bacillus subtilis* (Supplementary Movie 1), *Escherichia coli* (Supplementary

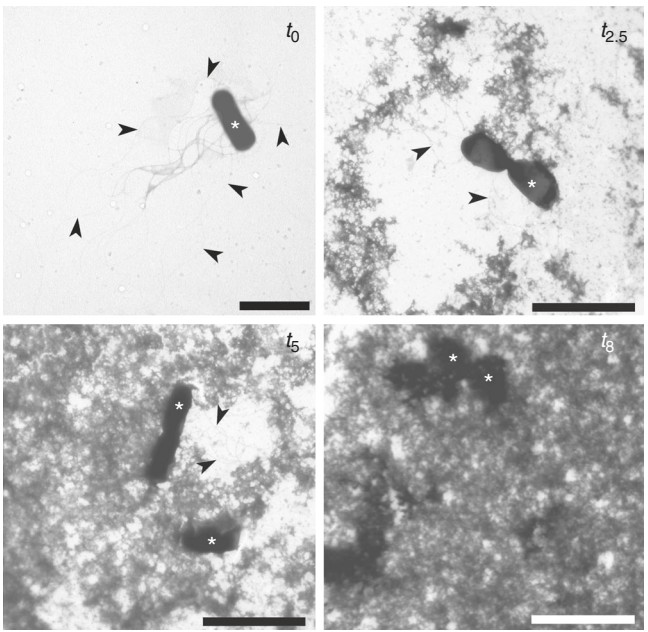

**Fig. 1** Development of the *B. subtilis* wt extracellular network. TEM micrographs of inoculated SM growth medium ($t_0$) and *B. subtilis* wt cultures after 2.5, 5, and 8 h of incubation. The extracellular material was partially attached to cells (*asterisk*), flagella (*arrow*) or was excreted in the extracellular space. Scale bars represent 2 μm for $t_0$ and 3 μm for $t_{2.5}$, $t_5$ and $t_8$

Movie 2), *Vibrio ruber* (Supplementary Movie 3), *Staphylococcus aureus* (Supplementary Movie 4), *Pseudomonas fluorescens* (Supplementary Movie 5), and *Pseudomonas stutzeri* (Supplementary Movie 6). Bacteria coupled in a cluster followed the motion of the optically trapped bacterium. The extent of coupling in different bacterial suspensions was very large and ranged from 60 to 140 μm. Mechanically coupled cells may sometimes form visible aggregates in suspensions. For example, *P. aeruginosa* cells produce non-attached aggregates during planktonic growth in the size range of 10–400 μm in diameter[20]. We have suspected that prior to visible aggregate formation *P. aeruginosa* cells are already mechanically coupled in the suspension. Stimulated by such a hypothesis we have prepared *P. aeruginosa* bacterial suspensions and checked for the mechanical coupling. As given in Supplementary Movie 7, *P. aeruginosa* cells were mechanically coupled before the formation of visible cell aggregates. This indicates that seemingly solo cells in the culture were in fact coupled prior to the macroscopic aggregation. Mechanical coupling was not apparent by bright field or contrast enhancing microscopy techniques.

**Early formation of extracellular matrix**. The formed extracellular matrix was visible in TEM micrographs. In *B. subtilis* (Fig. 1) the extracellular material was granular and formed fractal like structures that were partially attached to the cell surface and flagella. The electron density of the extracellular matrix material was comparable to the cell material. The amount of visible extracellular material increased exponentially, and most of the extracellular space was filled with the material after 8 h of incubation suggesting a fully interconnected network. To further characterize the extracellular material, we have used SEM microscopy (Supplementary Fig. 1), which indicates that extracellular material of *B. subtilis* may interact with the cell surface forming a smeared halo of material around the cell. Similar observations of hydrated EPS around individual

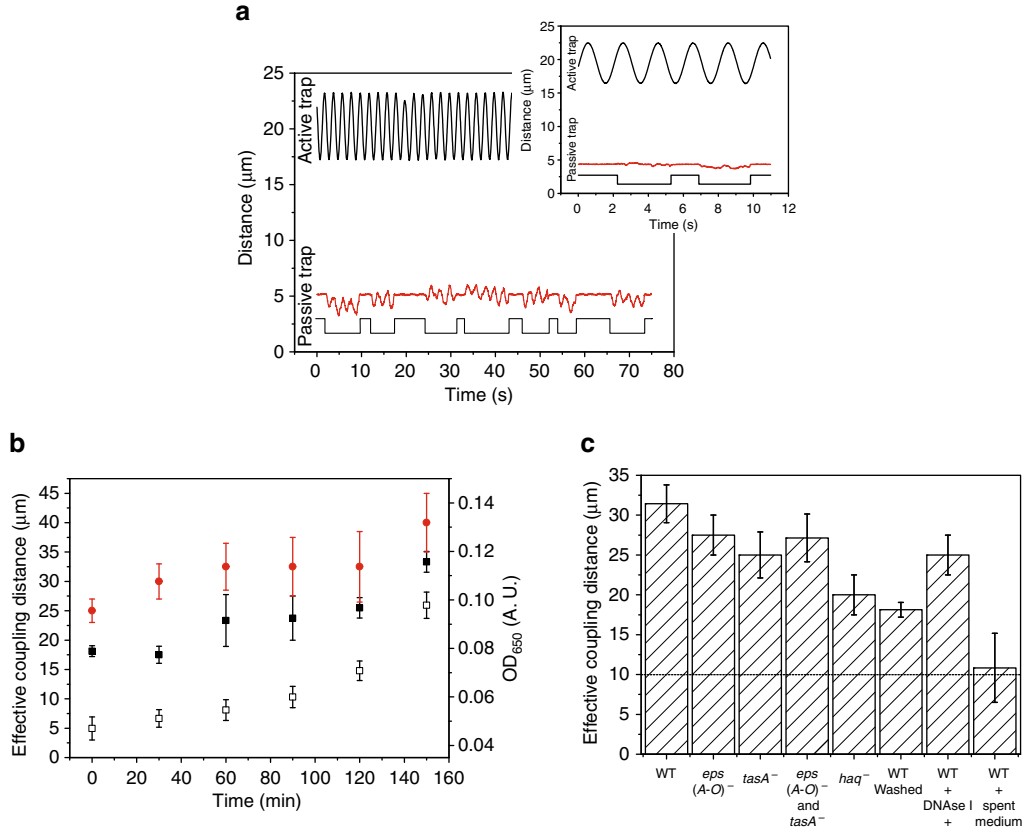

**Fig. 2** The effective mechanical coupling of optically trapped bacterial pairs. **a** A single cell was trapped in the active optical trap and oscillated longitudinally. Amplitude and frequency of oscillations were varied. At a distance $d$, a second bacterium was trapped and then released to follow the motion of oscillating bacterium. Occasionally, the bacterium escaped from the inactive passive trap and was brought back and released. Trajectories of the active (*black line*) and passive (*red line*) *B. subtilis* wt stationary cells washed, re-suspended and grown for 2.5 h are depicted ($d = 15\,\mu m$, active trap amplitude 3 $\mu m$, active trap frequency 0.5 Hz, passive trap amplitude 1 $\mu m$). Line below the passive trap trajectory depicts the state of the passive trap (active/inactive). The inset: trace of *B. subtilis* stationary cells washed, re-suspended and diluted in PBS recorded at the same conditions ($d = 15\,\mu m$, active trap amplitude 3 $\mu m$, active trap frequency 0.5 Hz, passive trap amplitude 1 $\mu m$). **b** The effective coupling distances between the active and passive *B. subtilis* bacterium during the incubation of washed and re-suspended stationary phase in SYM medium (*filled black squares*), exponentially grown cells that have been diluted and re-grown in SYM growth medium to the exponential phase three times before the optical tweezers measurements (*filled red circles*), and optical density of washed and re-suspended *B. subtilis* wt culture (*open black squares*) Average values and standard errors are given ($n = 8$ or more). **c** The effective coupling distances of *B. subtilis* wild type after 2.5 h of incubation, *eps(A-O)*⁻, *tasA*⁻, *eps(A-O)*⁻ and *tasA*⁻, as well as aflagellate *hag*⁻ *B. subtilis* mutants. Coupling distance of washed and re-suspended wild type cells in SYM medium, samples treated with DNAse I and proteinase K enzymes or samples treated with enzymes from the decaying biofilm are given. The maximal span of hydrodynamic coupling in simple solutions is indicated with the dashed line. Average values and standard errors are given ($n = 3$ or more)

*Schewanella oneidensis* cells have been made with cryo-TEM imagining[21]. On the other hand, in *E. coli* suspensions (Supplementary Fig. 2) which were less mechanically coupled the amount of the extracellular material was significantly lower. Cells formed clusters embedded in the extracellular matrix. The electron density of the extracellular material was different from the electron density of the cell material.

**Factors influencing mechanical coupling**. To better characterize the early mechanical coupling, we have developed an optical tweezers protocol with an active and passive optical trap between pairs of *B. subtilis* stationary cells washed, re-suspended in growth medium, and grown for 2.5 h (Fig. 2a). The passive bacterium moved in synchrony with the oscillating bacterium in dilute bacterial suspensions in the growth medium (Supplementary Movie 8), but not when a bacterial culture was washed, re-suspended, and diluted in PBS buffer (Fig. 2a inset). The amplitude of the passive bacterium decreased linearly with increasing distance of the active bacterium (Supplementary

Fig. 3). The extent of the mechanical coupling between bacterial pairs increased significantly during the incubation (Fig. 2b), consistently the strength of the coupling increased as well (Supplementary Table 1). In mature *B. subtilis* biofilms, the exopolysaccharides encoded by *eps(A-O)* and TasA extracellular proteins[22, 23] are required for biofilm formation. To check if these extracellular polymers contribute to the early mechanical coupling, the expression of P*ₑₚₛₐ-gfp* gene construct was monitored. The expression was low at low cell density, but increased significantly when cells reached the stationary growth phase (Supplementary Fig. 4). However, even in the stationary growth phase only a fraction of cells had a strong GFP expression (Supplementary Fig. 5) consistent with biphasic switch proposed by Chai et al.[24]. A subpopulation of cells strongly expressing GFP was visible after 26 h of incubation (Supplementary Fig. 6), which suggests that exopolysaccharides encoded by *eps(A-O)* and TasA extracellular proteins do not contribute significantly to the early mechanical coupling in bacterial suspensions. To confirm this, the mechanical coupling of *eps(A-O)*⁻ and *tasA*⁻ mutants

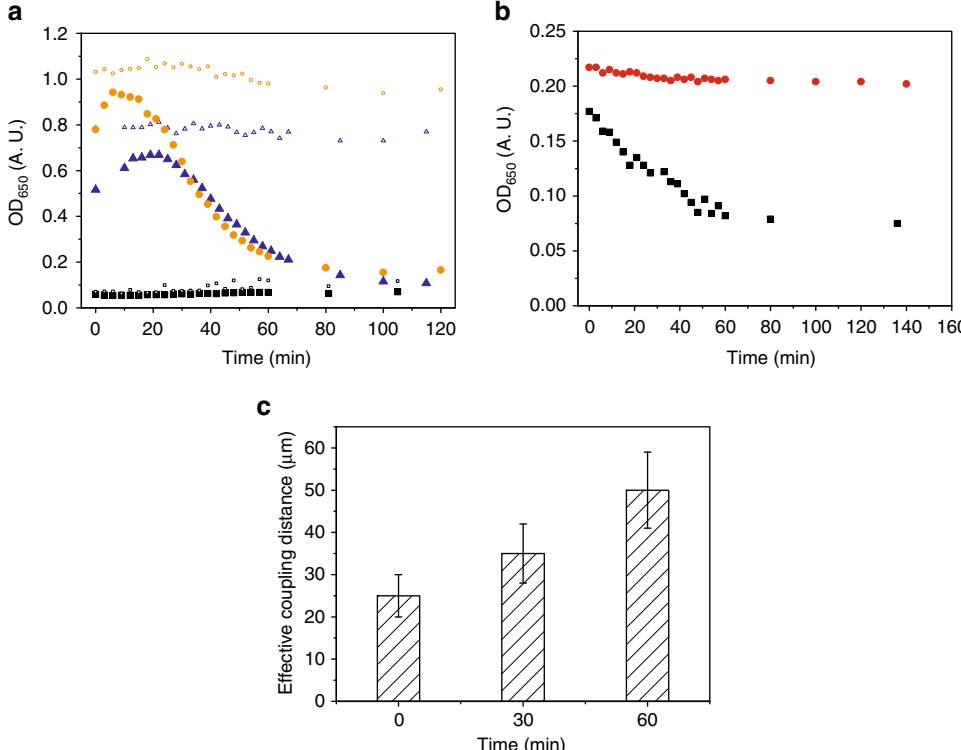

**Fig. 3** The effective mechanical coupling after cell lysis. **a** Cell lysis of exponentially grown *B. subtilis* wt cells in SYM at different initial optical densities at room temperature (*filled symbols*) or at 4 °C (*open symbols*). **b** Cell lysis of stationary (*filled red circles*) and exponential (*filled black squares*) *B. subtilis* cells re-suspended in PBS. **c** The effective mechanical coupling of *B. subtilis* wt cells during cell lysis. The average values and standard deviations are given (*n* = 3)

was measured at low cell density. The impact of *eps(A-O)⁻* or *tasA⁻* single or double mutant on the early coupling of bacteria cannot explain the observed coupling effect (Fig. 2c, Supplementary Fig. 7). In contrast to the mature biofilm[23], the early extracellular network did not contain measurable amounts of eDNA, the concentrations of reducing sugars were low (Supplementary Fig. 8). A small decrease in coupling was observed with the combined action of DNAse I and proteinase K (Fig. 2c). The coupling was mostly gone when spent medium from decaying biofilms was applied to dilute bacterial suspensions (Fig. 2c). The coupling could be affected by flagella. To check this, an aflagellate *hag⁻* mutant was tested. Although the coupling was still present, it decreased significantly (Fig. 2c). This suggests that the extracellular matrix may interact with the cell surface as well as with flagella. This has been further confirmed by TEM (Fig. 1). When an attempt was made to remove the unattached free EPS by washing, the coupling decreased significantly (Fig. 2c). If sodium azide, a respiration inhibitor, was added to bacteria at different times during the incubation, the coupling distance did not change significantly (Supplementary Fig. 9). This suggests that active bacteria are needed to build the extracellular matrix material. During the experiment bacteria divided a couple of times (Supplementary Fig. 10), which indicates that the mechanical coupling develops early in the growth phase. This is in sharp contrast to the currently accepted view that viscoelastic behavior typical for interconnected bacterial community only occurs at a transition into a stationary phase[19, 25, 26].

**Cell lysis contributes to mechanical coupling.** Although no visible extracellular material was present in stationary washed and re-suspended cells at the beginning of the experiment (Fig. 1), optical tweezers experiments indicated long-range interconnections between bacterial pairs (Fig. 2b). The long-range coupling

could be due to eDNA which has a role in the structure of the young biofilms but its function is later taken over by other exopolymers[27]. To check this, eDNA was visualized with TOTO-1 nanosensitive nucleic acid stain (Supplementary Fig. 11). The results indicate that the extracellular matrix of the stationary culture was infested with small fluorescence particles that swarmed in the intercellular space. In addition, fluorescence filaments interconnecting fluorescent cells were visible in the samples. If stationary culture was 100-fold diluted, the mechanical coupling between pairs of bacteria remained large (50 ± 5) µm. In sharp contrast, washing stationary cells and re-suspending them in the growth medium reduced the mechanical coupling to (18 ± 2) µm. No fluorescence particles or filaments were observed in the extracellular space of washed stationary culture. Less than 0.1 % of the washed and re-suspended cells were permeable for the nucleic acid stain.

To further minimize the effect of possible pre-seeded connections that could be transferred by cell washing and inoculation, we have re-grown the culture several times to the exponential growth phase to obtain truly exponentially grown bacteria. The data indicate that mechanical coupling is still present (Fig. 2b). The rate of the coupling increase was similar to the stationary cells which were washed, re-suspended, and grown for 2.5 h. Surprisingly, the coupling strength was higher in the exponentially grown cells. As exponentially grown *B. subtilis* cells are very sensitive to different environmental stress conditions[28], we have checked for cell lysis in unshaken bacterial cultures (Fig. 3a). Initially, bacterial cells at higher biomass densities continued to grow before the onset of a massive cell lysis. On the other hand, at low cell densities, similar to the conditions in the optical tweezers experiments, we did not observe cell lysis. Cell lysis was not pronounced when cells were cold shocked to 4 °C at either high or low cell density for a prolonged period of time. If exponential cells were washed and re-suspended in PBS buffer,

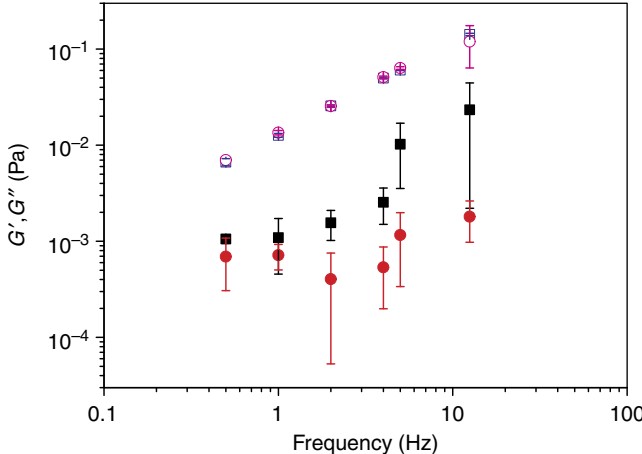

**Fig. 4** Viscoelastic properties of bacterial local environment. Storage modulus ($G'$, filled symbols) and loss modulus ($G''$, open symbols). Storage and loss moduli of SYM growth medium are represented with circles. The extracellular matrix storage and loss moduli of dilute *B. subtilis* wt bacterial suspensions are represented with squares. The average values and standard deviations are given ($n = 6$)

cells started to lyse immediately (Fig. 3b). This was different to the stationary cells that were washed and re-suspended in PBS, where cell lysis was much less pronounced. When stationary cells were incubated in PBS for 2.5 h coupling did not change significantly (Supplementary Fig. 12). At the end of the incubation in PBS the coupling was 30 μm, which was lower compared to cells incubated in the growth medium (Fig. 2b). This implies that increased coupling measured in the growth medium was due to the new production of the extracellular matrix material.

An indication that the exponential cells are more prone to cell lysis compared to the stationary, cells were further checked with TOTO-1 nucleic acid stain. Stained exponentially grown samples were not incubated for 15 min as recommended by the manufacturer, but were immediately taken for observation under the microscope. 2 min after sample preparation ~2 % of the exponential cells were intensively fluorescing indicating that cell membranes were compromised. This is approximately an order of magnitude higher than in the overnight bacterial suspension. With increasing time of microscopy more cells start to fluoresce (Supplementary Fig. 13). Already after 30 min of microscopic observations small fluorescent corpuscular bodies appeared in the vicinity of the dying bacterial cells. Small corpuscles eventually formed a halo of swarming fluorescence bodies around a cell. Most of the fluorescence bodies were tethered to the dying cell. A fraction of fluorescence corpuscular bodies moved freely in the medium. We have observed that dying cells were frequently interconnected with long fluorescence filaments not present at the beginning. Using SEM microscopy (Supplementary Fig. 14), one could observe a progressive morphological decay during cell lysis.

Cell division is an extremely complex process where a fraction of cells proportional to the growth rate will inevitably lyse[29]. The released cell material may contribute to the formation of the extracellular matrix material and consequently to the mechanical coupling of bacterial pairs. To demonstrate this, exponentially grown cells were lysed. The effective coupling distance increased significantly from ($25 \pm 3$) μm at the beginning to ($50 \pm 6$) μm after 60 min of cell lysis (Fig. 3c). This is a strong indication that lysed cell material contributes to the mechanical coupling in the extracellular matrix.

The cell lysis of exponentially grown cells was less pronounced or absent in other bacterial species (Supplementary Fig. 15). The

results indicate that optical density increased in *E. coli*, *V. ruber*, and *P. aeruginosa*, did not change in *P. fluorescens* or *P. stutzeri*, and slightly decreased in *S. aureus*. It is important to note that the mechanical coupling was present in bacterial suspensions also in the absence of massive cell lysis (Movies 2–7). The coupling was not necessarily weaker. For example, a rather strong coupling was observed in *P. aeruginosa* bacterial suspensions prior to visible aggregate formation.

**Low density bacterial local environment is viscoelastic.** Microrheology enables one to probe local viscoelastic environment of bacterial cells in situ in dilute bacterial suspensions. Active one-particle microrheology measurements were performed using a precise computer-controlled sinusoidal modulation of trapping-beam position and synchronous measurement of particle position. This makes it possible to determine storage and loss modulus, $G'$ and $G''$, respectively. The results (Fig. 4) indicate that the extracellular matrix material is viscoelastic in nature. The loss modulus was not significantly different from the medium. There was, however, a significant elastic contribution of the extracellular matrix that was not present in the growth medium. Both storage and loss modulus increased with increasing frequency. The frequency response observed was typical for the behavior of weak viscoelastic liquid composed of unlinked polymers[30].

Viscoelastic nature of the extracellular matrix material could explain the observed mechanical coupling between bacterial pairs. To check if the mechanical coupling was not due to the effect of hydrodynamic interaction in simple solutions (without macromolecules), we have systematically varied distance between the active and passive bacteria in dilute PBS suspension (cells were washed and diluted $10^4$-fold). The results indicate that the scale length of the detectable hydrodynamic forcing was less than 10 μm (Supplementary Fig. 16). If cells were replaced with smooth silica particles of similar dimensions, the extent of detectable hydrodynamic interactions was again less than 10 μm (Supplementary Fig. 17). The simple hydrodynamic effect therefore did not explain the large coupling observed in bacterial suspensions. However, in complex polymer solutions long-range hydrodynamic interactions with no intrinsic length scale may exist[31]. A plot of the passive bacterium amplitude response $A$ multiplied by the inter-bacterial distance $r$ as a function of the inter-bacterial distance $r$ indicates that a response $A$ is not strictly proportional to $1/r$ (Supplementary Fig. 18) suggesting that long-range coupling cannot be simply explained with the unscreened hydrodynamic effect.

## Discussion

Our work breaks with a central assumption that cells are not mechanically coupled with other bacteria in dilute bacterial suspensions. Though several bacteria have been described that form visible aggregates in liquid cultures (i.e., *Pseudomonas aeruginosa*[20, 32], *Sinorhizobium meliloti*[33], *Micrococcus luteus*[34], *Campylobacter jejuni*[35], *Streptococcus pyogenes*[36], *Staphylococcus aureus*[37]) the existence of mechanically coupled seemingly individual cells in dilute suspensions has not been convincingly demonstrated so far. We show that early viscoelastic structures in dilute bacterial suspensions are widespread in different Gram positive and Gram negative bacterial cultures. The strength of the mechanical coupling is proportional to the amount of the extracellular matrix material. Different type of polymeric material, both secreted as well as lysed cell material may contribute to the mechanical coupling. The formed extracellular matrix is a weak viscoelastic liquid which begins to form early in the exponential growth.

We have further shown that simply diluting the bacterial overnight cultures in the new growth medium by inoculation does more than just transferring cells to the new environment. Dilution decreases the mechanical coupling between bacterial pairs but the coupling may remain very large $(50 \pm 6)$ µm. A significant part of the coupling can be removed if cells are washed and re-suspended in the medium $(18 \pm 2)$ µm. The mechanical coupling grows over time both in the truly exponential or washed and re-suspended stationary cells suggesting that new material is constantly added to the extracellular matrix.

The long-range coupling of bacterial cells in dilute suspensions has not been observed before. In contrast, at high cell densities large scale cooperative effects have been demonstrated. For example, bacterial turbulence can be generated by collective swimming behavior of *B. subtilis* cells upon transition from random swimming to transient jet and vortex patterns in the bacteria/fluid mixtures at high cell densities (in excess of $10^9$ cells per ml)[38]. Such a collective swimming behavior reduces medium viscosity. We have not observed collective swimming behavior at low cell densities in *B. subtilis* suspensions. Nevertheless, in dilute bacterial suspensions cells may experience contact as well as hydrodynamic interactions with their neighbors. Recently, it has been demonstrated with optical tweezers that at short distances (<4 µm) contact interaction generates repulsive forces that prevent further cell–cell approach in *B. subtilis* cells[39]. The coupling interactions between pairs of optically trapped bacteria in dilute suspensions extend to much larger distances (up to 50 µm). The hydrodynamic effects measured in simple solutions (i.e., water or PBS) were less than 10 µm and cannot explain the observed mechanical coupling. The hydrodynamic properties of all but the simplest colloidal systems, however, are controversial and have been a subject of considerable debate[40]. A key factor in this uncertainty has been the intrinsically long-ranged nature of the hydrodynamic coupling between solid particles. It has been demonstrated for a pair of individual polymer-coated polymethylmethacrylate particles (diameter of 1.3 µm) using a passive microrheology that hydrodynamic coupling may extend up to 20 µm[31]. Bacterium with its numerous extensions (i.e., lipopolysaccharides, lipoproteins, pili, fimbria and flagella) is effectively a larger object than the cell body itself. Therefore, it is not entirely impossible that *B. subtilis* cells feel each other also hydrodynamically at distances of around 15 to 20 µm, which in most cases was the initially measured effective coupling distance.

As given by TEM and SEM micrographs, the extracellular matrix material may interact with the cell surface further increasing the apparent size of bacterium. In polymer solutions, dynamic properties of polymer chains affect hydrodynamic interactions and entanglements on large distances[41]. The existence of weak entangled polymer extracellular network is suggested by microrheology experiments performed at different modulation frequencies (Fig. 4). With increasing frequency, the extracellular polymer network is showing progressively more inflexibility and rigidity as the storage modulus increases more than the loss modulus[30] due to progressively reduced relative motion between the chains. As a consequence, elastic behavior $(G')$ is dominating over viscous behavior $(G'')$ with increasing frequencies and the two curves approach each other at higher frequencies. The measured viscoelastic effect is weak and swimming through such a network is not a problem for bacteria. However, if new biopolymer material is added it would fortify the network and eventually overcrowd the extracellular space as was observed at high cell densities (Fig. 1 and Supplementary Fig. 2). This will increase viscoelasticity of the extracellular matrix and will pave the way for biofilm formation. For example, the existing weak extracellular matrix in dilute bacterial *B. subtilis* cultures

could be reinforced with specific extracellular polymers (i.e., with polysaccharide encoded by *eps(A-O)* and TasA proteins) at high cell densities which may catalyze the coalescence of the network to biofilm pellicle formation. Similarly, in *P. aeruginosa* a high mechanical coupling observed already prior to aggregate formation in bacterial suspensions is conducive for phase transition to visible aggregates. It has been shown that a subpopulation of *P. aeruginosa* cells with increased type IV fimbriae production, defective in swimming, catalyze aggregate formation in bacterial suspensions[32]. Biofilm formation in bacterial suspension can be viewed as a phase transition process facilitated by a sudden release of polymer material. For example, it has been demonstrated recently that the explosive cell lysis of a sub-population of *P. aeruginosa* cells facilitates biofilm formation[42]. Similarly, we have shown that a sudden surge of DNA may induce a rapid phase transition in polysaccharide solutions[10].

Cell lysis also offers an explanation why live/dead assay regularly fails on exponentially grown *B. subtilis* cells, but give meaningful results in the stationary phase. We have repeatedly observed that the vast majority of the exponentially grown cells turn red in live/dead assay. The results of cell lysis experiments indicate that *B. subtilis* cell membranes may become compromised during the relatively short incubation period which is required according to the manufactures protocol for live/dead assay. Shaken exponential cells, on the other hand, continue to grow and reach the stationary phase, when cells are less sensitive to environmental perturbations. This explains why the majority of stationary cells are green with intact membranes after live/dead assay performed in the stationary phase.

Our mechanistic approach using optical tweezers proves to be very sensitive to local bacterial environment and enables one to follow the formation of the fragile extracellular viscoelastic network in dilute bacterial cultures. The method is robust as it is not critically dependent on the chemical details of polymers involved in mechanical coupling. With the new approach the sensitivity of the determination of plankton/non-plankton transition was shifted to much lower cell densities as currently accepted. The existence of the fine fabric of bacterial local environment at low cell densities is fundamental to bacterial physiology and will spur new experiments in the field. There are several unsolved issues in microbiology that are dependent on the understanding of bacterial local environment such as osmotic flow of nutrients through viscoelastic liquids, the onset and diffusion of signal molecules in quorum sensing, or different efficacy of antibiotic treatments at low and high bacterial densities[43] with a sharp transition around $10^7$ cells per ml. The discovery of the early viscoelastic extracellular matrix may provide a necessary framework for future experiments in the field.

## Methods

**Bacterial strains and growth.** In the optical tweezers experiments *Escherichia coli* MG 1655, *Pseudomonas aeruginosa* EXB V 129, *Pseudomonas fluorescens* CCM 2799, *Pseudomonas stutzeri* JM300 (DSM10701), *Staphylococcus aureus* sub. *S. aureus* EXB V 128, *Vibrio ruber* DSM 14379, and different *Bacillus subtilis* strains were used. All the bacterial strains except *B. subtilis* strains were obtained from the Ex Culture Collection of the Department of Biology, Biotechnical Faculty, University of Ljubljana (Infrastructural Centre Mycosmo, MRIC UL). The *Bacillus subtilis* NCIB 3610 wild type strain and its derivatives, the *tasA*⁻ single mutant (*tasA::spc*), *eps*⁻ single mutant (*eps(A-O)::tet*), the *tasA*⁻ *eps*⁻ double mutant (*tasA::spc, eps(A-O)::tet*)[22], and the *hag*⁻ single mutant (*hag::erm*, Ery^r) were generously provided by R. Kolter. Additionally, *B. subtilis* YC164 (P_epsA-*gfp* at the *amy* locus in NCIB 3610, Cm^r) strain was generously provided by Y. Chai and R. Losick[24]. All bacterial strains were stored at −80 °C. Prior to the experiments, strains were reactivated by culturing on Lysogeny broth agar plates (LB plates; 1.0 % (w/v) tryptone, 0.5 % (w/v) yeast extract, 1.0 % (w/v) NaCl, and 1.5 % (w/v) agar) at 28 °C for 48 h. In case of mutant strains, spectinomycin and/or tetracycline, erythromycin or chloramphenicol antibiotics were appropriately added into the LB agar in the final concentrations of 100 µg/ml of Sp and 20 µg/ml of Tc, Er, or Cm. Prior to the experiments, overnight (stationary phase) culture was grown in

LB growth medium without antibiotics at 28 °C at 200 r.p.m. for $(16 \pm 1)$ h. Stationary cells were washed by centrifugation at 10,000 g for 10 min (5424 Eppendorf, Hamburg, Germany) to separate cells from spent LB medium. After centrifugation, supernatant was discarded, an equal volume of saline solution (0.9 % (w/v) NaCl) was added, and the sample was vortex stirred for approximately 2 min. Cells were washed twice. After the third centrifugation, cells were vortex stirred in SYM medium, prepared as given by Shida et al.[44]: 70 mM $K_2HPO_4$, 30 mM $KH_2PO_4$, 25 mM $(NH_4)_2SO_4$, 0.5 mM $MgSO_4$, 0.01 mM $MnSO_4$, 22 mg/l ammonium iron (III) citrate, 2 % (w/v) yeast extract, and 20 % (w/v) sucrose. In 500 ml glass conical flasks with baffles that contained 100 ml of SYM medium, 2 % (v/v) inoculum was added. At regular intervals (0, 30, 60, 90, 120, and 150 min) samples were collected, their optical density at 650 nm was measured with MA 9510 photometer (Metrel, Brand, Germany), and samples were stored at 4 °C prior to optical tweezers experiments. Experimentally it was verified that storing samples at 4 °C did not change the mechanical coupling significantly. Unless otherwise stated stationary phase cells, washed and re-suspended in the fresh growth medium were used.

To obtain truly exponentially grown *B. subtilis*, 1% (v/v) inoculum of overnight culture was transferred directly to a fresh SYM medium. The bacterial culture was grown to the early exponential phase at $OD_{650} = 0.3$. Next, 1 % (v/v) inoculum of such culture was transferred to the fresh SYM medium. This has been repeated once more before the optical tweezers experiments.

To block respiration in *B. subtilis* and consequently to stop biosynthesis of new extracellular material, sodium azide was added to the stationary cells that were washed and re-suspended in SYM growth medium at final concentration of 7.7 mM (= 0.05 % (w/v)). Samples were incubated at 28 °C for 2.5 h. Sodium azide was also added to cultures which were growing for 1.0 or 2.0 h and were incubated in sodium azide for 1.5 or 0.5 h, respectively. To probe the effect of cell washing and dilution on mechanical coupling a $10^4$-fold diluted sample in SYM growth medium with 7.7 mM sodium azide was prepared.

**Cell density measurement**. At regular intervals (0, 30, 60, 90, 120, and 150 min) samples of *B. subtilis* wt were collected for cell density measurements. To immobilize cells for microscopy, $NaN_3$ (7.7 mM final concentration) was added. The DIC microscopy counts were obtained utilizing Axio Observer Z1 microscope (Zeiss, Göttingen, Germany) with 20 × objective (NA 0.4 and 1.6 × optovar, Zeiss) and Neubauer improved bright-line grid counting chamber with the depth of 25 µm (Brand, Germany). For each time point four chambers were manually evaluated by counting cells in the entire grid.

**Movies**. Movies used for mechanical coupling determination within bacterial clusters were recorded on an Eclipse Ti inverted microscope (Nikon, Tokyo, Japan) equipped with laser tweezers (Tweez 250si, Aresis, Ljubljana, Slovenia) and with a CMOS camera (UI-3370CP-M-GL, Obersulm, Germany) at approximately 50 frames per second. Bacterial cultures were grown to the exponential phase. *E. coli* was grown in LB at 37 °C, *S. aureus* in nutrient broth (0.3 % (w/v) beef extract, 0.5 % (w/v) peptone, 0.5 % (w/v) NaCl) at 37 °C, *P. aeruginosa* in nutrient broth at 28 °C, *P. fluorescens* and *P. stutzeri* in LB at 28 °C, *B. subtilis* in SYM at 28 °C, and *V. ruber* in PYE[45] (0.5 % (w/v) peptone, 0.1 % (w/v) yeast extract, 0.2 % (w/v) $MgCl_2 \times 6H_2O$, 0.3 % (w/v) NaCl) at 28 °C. Bacterial cultures were reactivated by first growing them on solid agar plates in their respective medium, overnight cultures were prepared in the same liquid media as used for reactivation and 1 % (v/v) inoculum was transferred to the 100 ml glass conical flasks containing 20 ml medium. Bacterial cultures were grown at 200 rpm to $OD_{650}$ of 0.5. Because *P. aeruginosa* formed visible aggregate in the liquid culture at $OD_{650}$ of 0.5 it was grown to $OD_{650}$ of 0.2 only when no visible aggregates were noticed. Samples were stored at 4 °C prior to movie recording. All recordings were done at room temperature. It was experimentally checked that storage at 4 °C did not induce the observed mechanical coupling effect. To stop cell motility, sodium azide was added to samples at final concentration of 7.7 mM. Optically trapped bacterium was moved by hand in the whole field of view at different speeds and directions to get a better visual impression of the viscoelastic coupling effect.

To estimate the mechanical coupling in bacterial clusters, optically trapped bacterium was sinusoidally modulated with a constant amplitude of 25 µm at modulation frequency of 0.2 or 1.0 Hz. Bacterial correlated motions were obtained by analyzing recorded movies. The reach of the correlated motion of bacteria in the cluster was estimated by measuring the maximal distance from the center of oscillating trapped bacterium at angles of 45° relative to the line of motion of the trapped bacterium. Trapping plane was set at least 80 µm from the walls to minimize surface effects. The bacterium showed the correlated motion within the bacterium cluster if it moved in phase with the trapped bacterium with amplitude of at least 1 µm to minimize the contribution of the Brownian motion. Care was taken to evaluate the reach only when a single bacterium was trapped, as the reach increased when several bacteria were simultaneously trapped and moved by the trap. The data were collected in different directions and average estimates were calculated.

**Determination of mechanical coupling between bacterial pairs**. The stiffness of optical trap was recalibrated before each set of measurements. We used a calibration method based on statistical analysis of bead motion in a stationary trap. At least 30,000 frames were recorded during each calibration run. Trap stiffness was determined by analyzing spatial distribution of trap positions using TweezPal software[46]. Correlated motion of the optically trapped bacterial pair was investigated using an inverted microscope (Nikon Eclipse Ti, Tokyo, Japan) equipped with laser tweezers (Tweez250si, Aresis, Ljubljana, Slovenia). An infrared laser beam with a wavelength of 1064 nm was focused through a water immersion objective (× 60, NA 1.00, Nikon) in a sample cell and was used for trapping and manipulation of bacteria. Sample cell with volume of approximately 50 µl was prepared with two cover slips ($60 \times 24$ mm², #1.5 and $20 \times 20$ mm², #1.5) separated by 200 µm spacers and sealed with silicone paste. Two optical traps were used to determine correlated motion of the trapped bacteria. The position of the first optical trap with the bacterium was sinusoidally modulated (an active trap) while the second bacterium was trapped in the stationary trap (a passive trap) in the direction of the line connecting the bacteria (longitudinal direction). Trapping plane was set at least 30 µm from bottom wall and at least 70 µm from the upper wall to minimize boundary effects. Next, the passive trap was switched off and the bacterium was allowed to follow the oscillations of the active trap. We have observed correlated motion in a moment when bacterium was released from the passive trap. When the passive bacterium drifted too far from the starting position, we shortly switched the passive trap on to reposition it. The correlated motion of two bacteria was measured at distances from 10 to 45 µm with a 5 µm increment between the two optical traps. The frequency and amplitude of the active trap was varied. After initial optimization, frequency and amplitude of the active trap were fixed at 0.5 Hz and 3 µm, respectively. Positions of the two bacteria were recorded with a CMOS camera (UI-3370CP-M-GL, Obersulm, Germany) at ~ 50 frames per second. The camera image acquisition was synchronized with trap movement using external camera trigger so that the phase lag between the bacteria and trap positions could be exactly determined. Bacterial trajectories were obtained by analyzing recorded videos with particle tracking software (PartTrack V3.36 for Aresis Tweez) and further analyzed utilizing Origin 9.0 (OriginLab, MA, USA). Limit of detection of passive trap amplitude was set at 0.20 µm. The maximal distance at which the amplitude of passive trap was still reliably detectable (an effective coupling distance) was determined by increasing the distance between the traps. After 5 h of incubation the coupling distance measurements were not determined due to the increased cell densities and overlapping concentrations of extracellular polymeric substances. Samples for optical tweezers experiments were stored at 4 °C to prevent cell lysis. All the measurements on optical tweezers have been done at room temperature. Storing the samples at 4 °C will inevitably decrease the metabolic and swimming activity. As determined by DIC microscopy cells did not aggregate at low temperatures. We have compared the mechanical coupling of the samples that have been at room temperature with sample that have been stored at low temperatures prior to the measurement. In both cases cells were mechanically coupled. The samples that have been at room temperature had slightly higher but not significant mechanical coupling compared to samples that were kept at low temperature for the same duration. Storing cells at low temperatures did not induce the viscoelastic effect observed.

**Microrheology measurements**. Microrheological experiments were performed utilizing an Eclipse Ti inverted microscope (Nikon, Tokyo, Japan) equipped with laser tweezers (Tweez250si, Aresis, Ljubljana, Slovenia) as described previously[10]. The position of the optical trap was sinusoidally modulated applying constant amplitude of 0.3 µm and frequencies of 0.5, 1, 2, 4, 5, and 12.5 Hz. Bead trajectories were obtained by analyzing recorded videos with particle-tracking software (PartTrack V3. 36, Aresis). Bead and laser-trap trajectories were further analyzed with custom-written analysis software in MatLab to obtain the microrheological parameters. Typically, 1 µl of the original bead suspension (radius $a = 2.32$ µm SS04N, Bangs Labs, Fishers, IN) was diluted with Millipore water (Billerica, MA) by a factor of $10^3$. Next, 1 ml of bacterial suspension was mixed with 60 µl of bead suspension. The storage and loss modulus were obtained as described previously[10].

**Hydrodynamic interactions**. In the dilute regime, the hydrodynamic interactions between pairs of bacteria (or passive particles) are often ignored as the interparticle distance exceeds the range of the flows resulting from the particle motion. With increasing bacterial cell density, however, the hydrodynamic effect can no longer be ignored. The subject of hydrodynamic interaction between two or more particles at low Reynolds number has been thoroughly reviewed[47]. Generally, hydrodynamic interaction between particles is governed by the following variables: shape, sizes, distance, orientations with respect to each other; individual orientations relative to the direction of the gravitational field, velocities, and spin relative to the fluid at infinity. Since the Reynolds number based on a typical size of the particles is small, the local fluid motion is assumed to satisfy the quasi-static Stokes equations. Due to the linearity of the governing equations of motions and boundary conditions, two modes of motion, translation and rotation, may be separately investigated, and the results superposed. In the case of optical tweezers, one can restrict only to the particle translation, without rotation. In the case of spherical particles, the force exerted by the fluid on each particle if the two particles

move along their line of centers is given by

$$\mathbf{F} = \frac{6\pi\mu\, a}{1 + (3/2)(a/d)} U \tag{1}$$

And if they move perpendicular to their line of centers by

$$\mathbf{F} = \frac{6\pi\mu\, a}{1 + (3/4)(a/d)} U \tag{2}$$

Where $\mathbf{F}$ is force, $\mu$ is kinematic viscosity, $a$ is radius of a particle, $d$ is distance between particles and $U$ instantaneous velocity. The hydrodynamic effect is linearly decreasing with separation distance and is approximately twice as strong along the line of centers compared to perpendicular direction. The hydrodynamic effect is expected to be important at a distance up to several diameters of particles. For example, the drag force by two swimming bacteria arranged head to tail was experienced up to 6 times of the total length of each bacterium[48]. In the case of *B. subtilis* wt, this would be up to approximately ~ 15 μm. To experimentally check for hydrodynamic interactions both *B. subtilis* wt bacteria and silica beads with a radius of $a = 2.32\,\mu m$ (Bangs Labs, SS04N) of approximately the size of the length of a bacterium were used. The correlated motion of the two silica beads was measured at distances from 7 to 15 μm between the two optical traps. Trapping plane was set at least 30 μm from the bottom surface to neglect surface effect. One of the particles was oscillated by an active trap (full stiffness) with amplitude of 3 μm and frequency of 0.5 Hz in the direction, which connects centers of the beads (longitudinal, direction). Probe bead was held at the fixed position in a weak trap, at lowest possible stiffness (20 % of full stiffness). Motions of the beads were recorded for 5 min. Fast Fourier Transform was applied to the data.

**Extracellular matrix digestion by DNAse I and proteinase K.** Reaction mixture containing 1 ml of *B. subtilis* wild type culture after 2.5 h of growth as described above, 7.7 mM NaN$_3$ final concentration (to prevent further bacterial growth), 5 units of DNAse I (Thermo Fisher Scientific, USA), and 100 μg/ml proteinase K (Sigma Aldrich, USA) was prepared. Proteinase K was added to the reaction mixture approximately 3 h after the addition of DNAse I to avoid possible enzymatic digestion of DNAse I by proteinase K. The samples were gently mixed by pipetting and left at 37 °C for $(17.5 \pm 0.5)$ h. The control reaction mixture without enzymes was also prepared. Samples were stored at 4 °C prior to optical tweezers experiments.

**Effect of spent medium on mechanical coupling.** *B. subtilis* NCIB 3610 wt culture was grown in MSgg minimal medium[49]. The composition of the MSgg was 5 mM potassium phosphate (pH 7), 100 mM MOPS (pH 7), 2 mM MgCl$_2$, 700 μM CaCl$_2$, 50 μM MnCl$_2$, 50 μM FeCl$_3$, 1 μM ZnCl$_2$, 2 μM thiamine, 0.5 % (v/v) glycerol, 0.5 % (w/v) glutamate, 50 μg/ml tryptophan, and 50 μg/ml phenylalanine. Cells were grown at 37 °C for $(53.5 \pm 0.5)$ h and the spent medium was collected and centrifuged at $10,000 \times g$ for 10 min in order to separate cells from the macromolecules and smaller molecules that may alter the effective mechanical coupling. The supernatant was mixed with an equal volume of *B. subtilis* bacterial suspension that was grown for 2.5 h. To prevent further bacterial growth in the mixture, 7.7 mM sodium azide in final concentration was added. The samples were gently mixed by pipetting and left at 37 °C for $(17.5 \pm 0.5)$ h. The control sample with addition of sterile medium to *B. subtilis* wild type bacterial culture was also prepared. Samples were stored at 4 °C prior to the optical tweezers experiments.

**Determination of extracellular DNA.** At regular intervals (0, 30, 60, 90, 120, and 150 min) samples of *B. subtilis* wt grown in SYM medium without yeast extract were collected and their optical density at 650 nm was measured. Samples were centrifuged at $10,000 \times g$ for 10 min to separate cells from the medium containing extracellular DNA (eDNA). Concentration of ds eDNA was determined at 260 nm utilizing Nanodrop 1000 spectrophotometer (Thermo Scientific, USA) assuming 1 absorbance unit equals 50 μg/ml dsDNA and corrected for the background.

To visualize eDNA, TOTO-1 iodide (ThermoFisher Scientific, Molecular probes, USA) nucleic acid stain was used in final concentration of 2 μM. The stock solution was prepared by 10-fold dilution of the original concentrate in PBS. To 50 μl of bacterial culture, 2 μl of stock solution was added and gently mixed. 10 μl was transferred to microscope slide, covered ($20 \times 20$ mm$^2$, 1.5 # cover slide), and sealed by a mixture consisting of non-bleaching vasaline, paraffine and lanoline. The prepared samples were immediately observed under the Axio Observer Z1 epifluorescence microscope (Zeiss, Göttingen, Germany) using × 100/1.40 NA objective, mercury HBO 100W lamp, 38HE filter set, and recorded with coupled MRm Axiocam camera (Zeiss).

**DNS assay.** At regular intervals (0, 30, 60, 90, 120, and 150 min) samples of *B. subtilis* wt in SYM medium were collected and their optical density at 650 nm was measured. Samples were centrifuged at $10,000 \times g$ for 10 min to separate cells from the growth medium. To 600 μl of supernatant, an equal volume of DNS reagent (5

g 3, 5-dinitrosalicylic acid and 150 g potassium sodium tartrate tetrahydrate dissolved in 100 ml 2 M NaOH and dH$_2$O added to 500 ml) was added into borosilicate vials. Samples with 0, 1, 2, 3, 4, and 5 mM glucose dissolved in SYM medium without sucrose were used to prepare a calibration curve. Vials were capped and heated to 100 °C for 15 min. The absorbance was measured using a plate reader (Multiskan Spectrum, Thermo Electron, Vantaa, Finland) at 575 nm.

**Transmission electron microscopy.** For negative staining transmission electron microscopy (TEM), ~ 20 μl of samples were transferred and left for 5 min on grids covered with Formvar support film (SPI supplies, USA) and negatively stained with 1 % (w/v) aqueous uranyl acetate for 5 s. *B. subtilis* and *E. coli* were grown in SM (SYM[44] without yeast extract) and M9 medium (6.4 % (w/v) Na$_2$HPO$_4 \times$ 7H$_2$O, 1.5 % (w/v) KH$_2$PO$_4$. 0.25 % (w/v) NaCl, 0.5 % (w/v) NH$_4$Cl, 2 mM MgSO$_4$, 0.4 % (w/v) glucose, 0.1 mM CaCl$_2$), respectively, to reduce the polymeric background otherwise present in SYM and LB media. The samples were fixed immediately to prevent further modifications. Excess staining solution was removed by a filter paper. Stained bacteria were air dried at room temperature and examined with a Philips CM 100 electron microscope at 80 keV.

**Scanning electron microscopy.** For scanning electron microscopy (SEM), the bacterial suspensions were fixed in 1% formaldehyde and 0.5% glutaraldehyde in a 0.1 M cacodylate buffer and applied to pre-cleaned glass slides. After rinsing in 0.1 M cacodylate buffer, the attached cells were postfixed in a 1% aqueous solution of osmium tetroxide for 1 h, rinsed with water and dehydrated in an ethanol series (50%, 70%, 90%, 96%, and 100%, for 5 min each). Dehydrated samples were transferred to acetone, which was gradually replaced with hexamethyldisilazane. The samples were air dried, attached to metal holders and sputter-coated with platinum. Prepared specimens were examined with a JSM-7500F field emission scanning electron microscope (JEOL).

**Determination of eps operon expression.** Green fluorescent protein (GFP) labeled *B. subtilis* YC164 (with P$_{epsA}$-*gfp* gene construct) and wild type strains were used to determine the expression of *eps* operon and autofluorescence, respectively. Stationary cultures of both strains were grown as described above for 2.5 h and centrifuged at $10,000 \times g$ for 10 min in order to separate cells from the SYM medium, which was replaced by an equal volume of phosphate-buffered saline solution (PBS) (8.00 g/l NaCl, 0.20 g/l KCl, 1.44 g/l Na$_2$HPO$_4$, and 0.24 g/l KH$_2$PO$_4$, pH 7.4). Cells were allowed to adhere to wells of the diagnostic slides (Gerhard Menzel GmbH, Germany), precoated with 0.05 % (w/v) poly L-lysine, for approximately 15 min. After rinsing with PBS, to remove the unattached cells and dispersing the excess fluid, the SlowFade Gold antifade reagent (Life technologies, Termo Fisher Scientific, USA) was added to reduce the photo-bleaching of GFP, and slides were covered with $60 \times 24$ mm$^2$, #1.5 cover slips. Slides were examined with Axio Observer Z1 epifluorescence microscope (Zeiss, Göttingen, Germany). Differential interference contrast (DIC) and fluorescence images were observed (objective × 100, $NA$ 1.4, Zeiss) and recorded with a coupled MRm Axiocam camera (Zeiss). For each time point at least 1000, but on average 5000 bacteria were analyzed for single cell fluorescence intensity with custom written script for ImageJ 1.48b. The white flat field correction was digitally achieved by the script, whereas the fluorescence flat field correction was done by normalization to sodium fluoresceinate standard (0.75 g of sodium fluoresceinate diluted in 1 ml of 0.1 M NaHCO$_3$). For individual cells, identified by DIC, cell fluorescence intensity normalized to sodium fluoresceinate standard was obtained from fluorescence images. By measuring single cells fluorescence with microscopy, we were able to determine the autofluorescence distributions of the wt strain (strain without GFP) and the fluorescence distribution of P$_{epsA}$-*gfp* labeled strain. In the latter strain, there are two contributions to the fluorescence: the first is the fluorescence of GFP itself and the second is the autofluorescence. To obtain the distribution of the fluorescence of GFP, one cannot simply subtract the background fluorescence from the fluorescence of P$_{epsA}$-*gfp* strain, because by doing so one would assume the two contributions are correlated i.e. cells with high expression of *gfp* have also high expression of other fluorochromes. Instead, we assumed the two contributions are independent. To obtain GFP fluorescence distribution, the spectra were deconvoluted. The best fit parameters that determine the lognormal GFP distribution were than used to plot the GFP distribution. Fitting was performed in OriginPro 9.0, all fits had $r^2 > 0.97$.

**Cell lysis experiments.** Bacterial cultures were grown to OD$_{650}$ of 0.5 at 200 r.p.m. in 100 ml glass conical flasks with baffles containing 20 ml of appropriate medium. *E. coli* and *S. aureus* were grown at 37 °C in LB and nutrient broth, respectively. The other bacteria were grown at 28 °C, *P. aeruginosa* in nutrient broth, *P. fluorescens* and *P. stutzeri* in LB, *B. subtilis* in SYM and *V. ruber* in PYE. Because *P. aeruginosa* formed visible aggregates in the liquid culture at higher optical densities, growth was stopped at OD$_{650}$ of 0.2 when no visible aggregates were noticed. Next, 2 ml of the exponentially grown culture was transferred to cuvette and incubated unshaken at room temperature. OD$_{650}$ was measured in regular intervals with MA 9510 photometer (Metrel, Brand, Germany). To check the effect of initial cell density on cell lysis, *B. subtilis* cells were grown from the early to mid-exponential phase, when cells were transferred to cuvette and incubated at

room temperature unshaken with regular measurement of $OD_{650}$. To check the effect of PBS on cell lysis, either cells that were incubated overnight or truly exponentially grown cells were washed twice in saline solution and re-suspended in PBS. Cells were transferred to cuvette and incubated at room temperature unshaken with regular $OD_{650}$ measurements.

**Data availability**. The authors declare that the relevant data supporting the findings of this study are available in the article and its Supplementary Information files, or from the corresponding author upon request.

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

## Acknowledgements

We thank Tjaša Danevčič and Natan Osterman for their contribution with initial optical tweezers experiments. This work has been supported by the European Commission under grants 211800 SBMPS, FP7/2007–2013, Slovenian Research Agency under grant J4-7637 (D) and University infrastructural center "Microscopy of biological samples" located in Biotechnical faculty, University of Ljubljana.

## Author contributions

D.S. conceived the project with the input from other authors. S.S. and B.S. prepared the samples for optical tweezers, B.S. and I.P. carried out the optical tweezers experiments. R.K. carried out the TEM and SEM experiments and prepared micrographs. I.D. performed cell density measurements, prepared custom written script for analysis of the *eps* operon expression and performed fluorescence microscopy lysis experiments. S.S. carried out bacterial growth experiments, enzymatic digestion, and fluorescent microscopy *eps* expression experiments. All authors contributed to the analysis and interpretation of the results. D.S. was responsible for the overall project strategy and management and wrote the manuscript with the help of S.S. and inputs from the other authors.

## Additional information

**Competing interests:** The authors declare no competing financial interests.

