## [Peer Review File · Nature Communications]

Reviewers' comments:

Reviewer #1 (Remarks to the Author):

General comments:

In the present study the authors have use an innovative technique of optical tweezers to determine if and to what degree bacterial cells are mechanically connected in a liquid batch culture. The study aims to answer the important question; are liquid bacterial culture really "planktonic". They find that seemingly solo single cells in a culture are in fact coupled in a mechanical network.

The consistency of liquid culture is indeed important to study as they appear much more complex than most microbiologists is aware of. However Planktonic does not mean single cells as it is very common for some microbiologist to define it. Planktonic just means passively floating or drifting around. This means that large non-attached biofilm aggregates may be just at planktonic as single cells. That said most liquid batch cultures are very far from the homogeneous mix of single cells of often thought of. Schleheck et al. 2009 finds large amounts of non-attached aggregates in liquid batch cultures of *Pseudomonas aeruginosa* after few hour of growth.

The presented technic of optical tweezers is indeed interesting and quite well explained. But the microbial aspects of the study are in my opinion poorly executed and introduce several immense uncertainties. First of all based on the description giving in materials and method, the authors are unfortunately mostly investigating compacted or attached cells from the overnight culture and not the development of connection in the experiment itself. This is unfortunate as the experiments are already "seeded" with connections before the experiment starts on its own. This is what is discussed in line 214-216. So basically the authors study dilute stationary cultures with a lot of matrix components. The washing etc is not enough to remove eDNA and other matrix components

Over all study lacks proper execution. It seems like the authors have focused more on the technical aspects of the method development than on the microbiology that they are investigating. Therefore I cannot support their conclusions based on this study.

Major points:

Materials and Method

It is hard to follow the logic behind selection of the investigated bacterial species, some of these are quite exotic, *P.stutzeri*, *V. rubber*. Why haven't the authors investigated the obvious choice of *Pseudomonas aeruginosa* or *Staph aureus* or...?

I find the growth and preparation of the cultures problematic. When the ON is centrifuged before inoculum, it will be expected that clumping and contact between cells will accrue in the inoculum when the biomass are compressed. This type of aggregation cannot simply be removed by vortexing. Therefore, there will be large amounts of pre-connected biomass which inoculated into the cultures at 0 h. This will constitute a quite large population as the authors inoculate with 2% ON into the experimental culture of 100ml. Potential you will have large amounts of pre-aggregated cells in the culture even before the experiment starts. This issue will embed itself in all experiments downstream. If the inoculum had been diluted, re-grown to exponential growth several times a quite uniform culture of single cells could have been achieved. I agree, that you see an increase in coupling over the 150 min, but if the initial connections are preformed, the premise of the study is missing.

When sampling the cultures the authors store the samples at 4°C until evaluation by optical tweezers. This is problematic as most bacteria will aggregate and precipitate when chilled in static conditions. This will add yet another possible false positive to the setup. I miss control experiments of the influence on aggregation and coupling over time, by the addition of both Sodium azide and

NaN₃ (figure 1e. only shows the effect of Sodium azide at 0 h).

The section on enzymes puzzle me. What are those enzymes the authors want to test? If the authors want to test the effect of spend media on the cultures, why don't they sterile filter the media? The effect of the spend media could be linked to a variety molecules into the medium; fatty acid, QS molecules, and other metabolic products.

Results:

I am missing the results *P.stutzeri*, *V. ruber*, *E.coli*. at least in Supplementary Materials.

Figure 2. It is very hard to identify flagella in the mist of dehydrated exo-polysaccharide. Are the authors positive that what they mark as flagella are these? In the top left panel, there would in this case be very large amounts of flagella. Flagella anti-bodies coupled with gold particles would be improve the detection.

Discussion:

Line 199 There are in fact quite a lot of studies published on non-attached aggregated biofilm in liquids.

Line 214-216 This sentence really summarizes the biggest problem with the study, the authors a are studying the formation of connections, but are studying whatever is transferred from the ON.

Line 217-219. There are no basis for the statement that growth rate should be affected in new cultures based on the current study, and no reference which can substantiate this are provided. The authors hypothesize that the formation of mechanically coupling may decrease the effect of antibiotics do to extra cellular crowding at high cell densities. This would be such a simple experiment, why was it not performed? The hypothesis of mechanically coupling may decrease the effect of antibiotics does not take any metabolic factors into account. It is well know that metabolic activity highly influence the affectivity of most types of antibiotics. In cultures with increased cell density, metabolic activity is most likely to decrease rapidly as well. I do not find any base in this study that underpins this direct coupling could explain an increased antibiotic tolerance, which cannot also be explained by shifts in metabolic activity.

Minor points:

Line 38 Please support statements with reference

Line 89 It is called *Vibrio ruber*

Line 139-142 Please support statements with reference

Line 212-214 Please support statements with reference

Line 229-231 revice the structure of sentence

Line 292 LB are not albeit for Luria-Bertani, but Lysogeny broth.

Line 292 Why are all strains grown at 28°C? *E.coli* optimal are 37°C, *Bacillus subtilis* are reported to be in the range 34-37°C. *Vibrio ruber* at 40°C.

Line 308-310 The authors fails to explain the rationale behind the addition of sodium azide. Why blocking the respiration?

Line 468 Temp. written in parentheses

Figure 1c r²-value of fitted model would be beneficial

Reviewer #2 (Remarks to the Author):

The article by Sretenovic et al. is concerned with the emergence of extracellular matrix coupling mechanically bacteria during the early growth of a planktonic population. Using TEM micrograph, they witness extracellular material attached to the flagella and the cellular body, even the early phase of the growth. The central claim of the paper is that they are able to measure a mechanical response due to this very tiny network which can significantly couple bacteria over large distances. From these measurements, they extract an interaction distance that they show to be growing with time as the bacteria concentration increases. They relate the growth of an interaction length with the existence of intercellular communication and quorum sensing. They claim that when the culture has reached a so-called “percolation threshold” that they define through a model, the communication is switched on. The author’s final claim is that this mechanistic explanation can solve a long standing problem of sudden resistance of bacterial populations to antibiotics.

This is a - priori an interesting paper, from the biological point of view the questions are relevant and well explained in the text. I agree with the authors on the crucial importance of the result if their claim is true. The author present a lot of data and follow a clear reasoning essentially based on the existence of a coupling mechanical length appearing during the early phase of the growth process. They present a model to correlate this length to the emergence of extra-cellular matrix connecting mechanically the bacteria. However due to the importance of the problem and also the high level of the journal they want to publish in, it also requires some high level of exigency on the significance of the measurements. There are many points that I do not understand, some on which I am skeptical and even some points I completely disagree with. I would be extremely pleased if the authors could reply precisely and convincingly to these remarks. At the present state, I do not think this paper is publishable in Nature Communications.

Let us review the different elements of proof.

1) The material existence of an extracellular matrix.

The micrographs of Fig. 2 seem indeed to show the existence of extra cellular matrix growing with time and especially at early time (2.5 h) one can observe some deposit. For me this is a central point which seems irrefutable provided no artifact is coming from the method. I do not see why it is not coming earlier in the exposition of arguments. I am not sure of the literature of the subject but is it something original from this paper to witness such extracellular matrix at such early time of growth? From the micrograph is it possible to estimate the volume content of this extracellular material? A related question is on what we see. Is the image presented statistically characteristic of the status at any point on the sample, or did the author have chosen some particular place especially eloquent to push their point?

2) Collective hydrodynamics dismissed or not?

The first thing to dismiss is the possibility of hydrodynamic coupling associated with the early existence of collective swimming of the bacteria that provides non-standard hydrodynamic coupling eventually reinforcing the response to a local perturbation and being an effect very different from the present interpretation. The authors quote the work of Gachelin et al (ref.17) on rheology. Note that in the same group there was recently measurements that may be pertinent to discuss along with these results. Lopez et al. (Lopez et al Phys. Rev. Lett. **115**, 028301 (2015)) show that classical rheology can be done at much lower shear rate (0.1 s^{-1}) that what is presented here and indeed these authors have shown a very peculiar rheological response for a sheared planktonic suspension of E.coli for about the same density range. Note that in their case, they use a fresh motility buffer to re-suspend the bacteria which in principle is cleaned from any extracellular matrix.

- a) First the statement that classical rheology method cannot access the suspension's rheology is not exactly true as shown by Lopez et al.
- b) Second, I think that the measurements presented in fig. 1.a, are useless essentially because they are not done in the right shear rate regime (around 100 s^{-1} !). At this shear rate basically any fine interconnected structure is likely to be destroyed and more importantly, the method is not adapted as to be able to probe viscoelastic response. In this perspective one should do oscillatory or time step response. I will come back to that but I do not see the interest in the main part of the text of such a rheological measurement obviously non-adapted to the question.

Sretenovic et al. use a PBS solution which in principle should get rid of extracellular matrix as in Lopez et al. and see a much weaker if any, long range hydrodynamic coupling.

- c) What is the guaranty that the bacteria are still active in this PBS medium? Is there some possibility to characterize the swimming activity by assessing the mean swimming velocity when these measurements are done?

Along those line, Gachelin et al (New Journal of Physics, **16**, 025003 (2014) have measured a correlation length coming from collective swimming of the bacteria and which grows linearly with the bacteria volume fraction. Therefore, collective swimming effect inducing hydrodynamic coupling may be important as well, along with the growth of an extracellular matrix.

- d) Is there a possibility to "kill" or asphyxiate the bacteria in the middle of the growth phase (and not at the beginning as in Fig.1e) to try to separate collective swimming hydrodynamics from polymeric matrix effects?

3) Measurement method to extract a coupling length

I agree with the authors that the local rheology measurement are a very useful tool to assess the mechanical coupling in the suspension. As I already discussed, oscillatory response is indeed a good way to probe viscoelastic coupling. I am convinced from the data presented by the authors that something indeed is going on when the concentration is increased. My main concern is on the methodology and on the analysis of the data allowing to extract a meaningful "typical length" representing a mechanical coupling. This is a central element that determines non-only the pertinence of this article and also the subsequent discussions in the manuscript based on the physical significance of this length.

From the oscillatory response, what the authors essentially assess is the intensity of the coupling (displacement) with distance. I have two major critics to do.

- a) From what I see for example in fig 1c, there is no strong argument in favor of a typical coupling length at least the way it is extracted from a linear fit. To illustrate this essential rebuttal, I have plotted the passive bacterium amplitude response R multiplied by the inter-bacteria distance r as a function of the inter-bacteria distance r . From what is seen in the subsequent figure, a response as $R \propto 1/r$, meaning essentially an effective hydrodynamic coupling in form of a power law decay (and no typical length scale) , is not at all impossible.

Data $R*r$ versus r extracted from fig. 1c.

- b) The second question is why the authors do not assess the phase response and do not vary the oscillation frequency to really test a viscoelastic signature? This is indeed a standard way to prove a visco-elastic coupling. Note that in this case, there is no clear length scale coming out from such a model.

I think that if the authors really want to push forwards convincingly the existence of a coupling distance (for whatever it means) they have to push the measurements over larger spanning distances so that one can really separate a response producing a convincing length scale from any power law decay. If the authors want to push forward the existence of an effective viscoelastic coupling growing with concentration, my impression is that they have to do it in the standard way and extract from the frequency response the viscous and elastic parameters.

4) Percolation model

Based on the assessment of a coupling length, the authors put forwards a percolation model to interpret their biological data and try to prove an onset of communication between the bacteria once the percolation threshold is reached. Such a relation between a topological transition and a global onset of communication inducing a quorum sensing response would indeed be very important and novel conceptually. I think this is the very weak point of the paper. Even with my previous doubts on the real existence of a length, the static percolation picture they present is hard to buy with swimming bacteria that would carry along their path this “corona” of extracellular matrix. I do not see in the data any convincing element proving the validity of such a picture. Moreover, I do not see any clear and convincing relation between a percolation threshold, even defined as the authors do, and the emergence of a quorum sensing signal.

Reviewer #3 (Remarks to the Author):

This study reports the observation of mechanical coupling between bacteria at nominally rather low densities due to extracellular substances. This is a very interesting observation that could indicate that our thinking about dilute bacterial suspensions is mostly wrong and could thus potentially be highly influential. However, extraordinary claims need extraordinary evidence and I am not sure that the required threshold of evidence is reached here; the importance of the results could also be much smaller (merely indicating that experiments must be done more carefully than what is usually done). In my opinion, the authors show interesting data in support of an exciting hypothesis, but not up to the point where they convinced me. Partly this is due to how the paper is presented (the small number of control experiments is shown very late), partly this is due to the lack of control experiments. In my opinion, the paper could be interesting enough to be considered for publication, but substantial additional work would be needed.

1) The first qualitative experiments (suppl. movies) did not convince me. Why is a different bacterium (*Pseudomonas*) used here than in the rest of the paper? The authors claim that they did this experiment with different species, but do not show any data.

Moreover, the authors claim to see a weak viscoelasticity of their suspensions when using optical tweezers, but not with other methods. They explain this by the higher sensitivity of optical tweezers, but to any non-expert in optical tweezers, an alternative explanation would be that this is an artefact of the method. I think any control experiment that shows a dependence on the medium (as the authors do later, e.g. replacing the medium with PBS or digestion of extracellular substance) would help a lot here, as it would show that the effect is specific to the bacterial cultures and not argue against artefacts. Ideally this would be accompanied by EM images of the suspensions.

In general, I think the authors need to convince the reader first that the effect they see is real. The present version of the manuscript is written from the standpoint that this effect exists and can be further studied. I think a change of standpoint towards establishing the existence of the effect and convincing the reader would be helpful. The authors say on p.3: "The results imply that different bacterial species are able to form cohesive viscoelastic networks in the dilute monoculture bacterial suspensions with persistence length of up to 40 μm ." At this point, this is a speculation, and cannot be concluded yet.

2) In the active-passive trapping experiment (Fig. 1), a control would be needed as well. The authors do a control experiment (Bacteria in PBS buffer instead of growth medium), but they present the data in a completely different form, so no comparison is possible. Why not show some traces as in Fig. 1b for correlated motion at different distances in SYN medium and for uncorrelated motion in PBS? Can an upper limit for the amplitude be estimated for the PBS experiment?

Likewise, show the Fourier transforms for both cases as well, so a direct comparison is possible.

3) In general I am wondering whether the bacteria in these experiments are actually in exponential growth phase or rather in lag phase. It seems what is considered here is the first generation (or the first two) after inoculation with some fraction of the extracellular substance responsible for viscoelasticity already present at inoculation. I would suggest a control experiment with cells that are truly in exponential phase due to repeated dilution at low OD with fresh medium. I would expect that the viscoelasticity is much lower then. This could weaken the claim of the authors, but in addition show that one has to be careful how a "dilute suspension" is generated. On the other hand, if the effect is observed in true exponential phase, the claim of the authors would be much stronger.

For the optical tweezer experiments, cells are stored at 4°C. Does this affect the observations? Is the same coupling seen if cells are directly moved to the tweezers?

4) In Fig 3, panel d is the most important one in my opinion. This panel contains some crucial control experiments. Specifically, for washed cells and spent medium, supporting the observation reported here. In addition, it provides the first hints on the genetic basis of the effect showing importance of flagella and lack of importance of eps and tasA.

The discussion of quorum sensing is not very convincing. The authors argue that quorum sensing sets in much later than percolation based on extracellular polysaccharide formation. I would find it more convincing if a quorum sensing reporter had been used that is independent of anything that affects viscoelasticity, but maybe this part of the manuscript can be rephrased with quorum sensing being more of an afterthought on an experiment done for other reasons.

Point by point reply

Reviewer #1 (Remarks to the Author):

General comments:

In the present study the authors have use an innovative technique of optical tweezers to determine if and to what degree bacterial cells are mechanically connected in a liquid batch culture. The study aims to answer the important question; are liquid bacterial culture really “planktonic”. They find that seemingly solo single cells in a culture are in fact coupled in a mechanical network.

*The consistency of liquid culture is indeed important to study as they appear much more complex than most microbiologists is aware of. However Planktonic does not mean single cells as it is very common for some microbiologist to define it. Planktonic just means passively floating or drifting around. This means that large non-attached biofilm aggregates may be just at planktonic as single cells. That said most liquid batch cultures are very far from the homogeneous mix of single cells of often thought of. Schleheck et al. 2009 finds large amounts of non-attached aggregates in liquid batch cultures of *Pseudomonas aeruginosa* after few hour of growth.*

The planktonic state is indeed ill defined in the literature. There is a wide distribution of different structures in planktonic bacterial suspensions ranging form single cells to different size cell aggregates. For example, as noticed by the reviewer *Pseudomonas aeruginosa* produces visible non-attached aggregates in the size range of 10–400 μm in diameter during the growth phase. We have suspected that even prior to visible aggregate formation *P. aeruginosa* cells were already mechanically coupled in the suspension. Stimulated by such a hypothesis, we have prepared *P. aeruginosa* bacterial suspensions and checked for the mechanical coupling. As given in **Supplementary Movie 7**, *P. aeruginosa* cells were mechanically coupled before the formation of visible cell aggregates. This indicates that seemingly solo cells in bacterial suspensions may couple prior to the macroscopic aggregation.

The revised text has been modified on p.3, l. 94-103.

The presented technic of optical tweezers is indeed interesting and quite well explained. But the microbial aspects of the study are in my opinion poorly executed and introduce several immense uncertainties. First of all based on the description giving in materials and method, the authors are unfortunately mostly investigating compacted or attached cells from the overnight culture and not the development of connection in the experiment itself. This is unfortunate as the experiments are already “seeded” with connections before the experiment starts on its own. This is what is discussed in line 214-216. So basically the authors study dilute stationary cultures with a lot of matrix components. The washing etc is not enough to remove eDNA and other matrix components.

This is the central issue on which we have spent most of the time preparing the revised manuscript. We have done new experiments to avoid already seeded connections at the beginning of the experiment. The inoculum has been diluted, re-grown to exponential growth several times and a uniform culture of single cells has been prepared prior to the optical tweezers experiments as suggested by the reviewer. We have checked for the presence of eDNA with nanosensitive TOTO-1 fluorescence microscopy. We have checked for the mechanical connections in the undiluted and diluted overnight cultures. We have checked for the effect of washing, cell lysis and growth media. The new results indicate that connections between cells were present in the truly exponential cells and that the strength of the coupling increased with time of incubation. The new experimental evidence will be discussed below.

Over all study lacks proper execution. It seems like the authors have focused more on the technical aspects of the method development than on the microbiology that they are investigating. Therefore I cannot support their conclusions based on this study.

We agree that the focus was on technical aspects of the method development which allowed us to see the new phenomenon for the first time. In the revised manuscript we have put much more effort on the microbiology. The new bacterial species were introduced. The effect of the physiological state of the bacterial cells (stationary vs. truly exponential cells) was investigated. The effect of cell lysis on the mechanical coupling was thoroughly studied. We have determined *in situ* viscoelasticity of bacterial local environment and have done several new control experiments to avoid artifacts. With the new experimental evidence we are reassured that the mechanical coupling is a widespread long-range phenomenon that exists in different bacterial dilute cultures, growth conditions, and methods of cell suspension preparation as will be presented and discussed below.

Major points:

Materials and Method

It is hard to follow the logic behind selection of the investigated bacterial species, some of these are quite exotic, P.stutzeri, V. ruber. Why haven't the authors investigated the obvious choice of Pseudomonas aeruginosa or Staph aureus or...?

The phenomenon of mechanically coupled cells in dilute planktonic state has not been described earlier. To determine how widespread the phenomenon is, we have used several bacterial species that we regularly cultivate in the lab. In the revised manuscript we have extended the list and included also *Pseudomonas aeruginosa* and *Staphylococcus aureus*. We now show movies of all tested bacteria. The recorded movies clearly convey the viscoelastic nature of the extracellular matrix and provide strong and unbiased evidence that convince the potential reader to the existence of the invisible network in different planktonic bacterial cultures. In addition, the reach of the mechanical coupling in different bacterial strains has been estimated. For further work we have

selected *B. subtilis*, a model bacterium in the lab, with which we have the most experience.

The text describing the movies has been rewritten on p.3, l. 87-93.

I find the growth and preparation of the cultures problematic. When the ON is centrifuged before inoculum, it will be expected that clumping and contact between cells will accrue in the inoculum when the biomass are compressed. This type of aggregation cannot simply be removed by vortexing. Therefore, there will be large amounts of pre-connected biomass which inoculated into the cultures at 0 h. This will constitute a quite large population as the authors inoculate with 2% ON into the experimental culture of 100ml. Potential you will have large amounts of pre-aggregated cells in the culture even before the experiment starts. This issue will embed itself in all experiments downstream. If the inoculum had been diluted, re-grown to exponential growth several times a quite uniform culture of single cells could have been achieved. I agree, that you see an increase in coupling over the 150 min, but if the initial connections are preformed, the premise of the study is missing.

This is the central issue on which we have spent most of the time preparing the revised version. Cell washing is used as a standard procedure in most microbiology labs, but as noted has a major drawback that cells are forcefully aggregated during the centrifugation step. As cells compact during the process, it is possible that extracellular matrix is concentrated in the interstitial volume and attaches irreversibly to the cell surface. Although cells were vigorously vortex mixed upon re-suspension, we checked for possible cell-cell aggregation with light microscopy. The aggregates were absent. In addition we performed tests to observe the efficiency of washing to remove the extracellular material. We checked for the presence of eDNA in the stationary cultures that were washed and re-suspended in the growth medium. The presence of eDNA in washed cells was determined with TOTO-1 nanosensitive nucleic acid stain. The results indicate that less than 0.1 % of cells were permeable for the nucleic stain. No fluorescence particles were present in the re-suspended samples and no fluorescence filaments were attached to the cells (**Supplementary Fig. 11**). In addition, we have prepared washed and re-suspended samples for TEM microscopy (**Fig. 1, Supplementary Fig. 2**). The micrographs indicate regular cells with no indication of aggregated extracellular matrix present in the medium or on the surface of the cells. In contrast, the extracellular matrix of non-washed overnight culture was infested with small fluorescence particles that swarm in the intercellular space (**Supplementary Fig. 11**). Some fluorescence filaments interconnecting fluorescent cells were visible in the samples. Most of the stationary cells were intact and impermeable to the nucleic stain. If stationary cultures were 100 fold diluted, the mechanical coupling between pairs of bacteria remained large (50 ± 5) μm . In sharp contrast washing stationary cells and re-suspending them in the growth medium reduced the mechanical coupling to (18 ± 2) μm suggesting that washing of cells was efficient.

The text describing eDNA has been added to the Result section on p. 6, l. 167-181. TOTO-1 staining procedure is described on p. 17, l. 598-606.

Although no visible extracellular material was present in washed and re-suspended stationary cells, optical tweezers experiments indicated long-range interconnections (i.e. 18 μm). To further minimize the effect of possible pre-seeded connections, we have as suggested re-grown the inoculum several times to the exponential growth phase (three times to $\text{OD}_{650} = 0.3$) prior to the optical tweezers experiments to obtain truly exponentially grown bacteria. In addition, we have used lower inoculum size (1%) in all subsequent experiments. The results for the mechanical coupling are presented in **Fig. 2b**. The data suggest that mechanical coupling is present also in the truly exponentially grown cells. Similarly to washed and re-suspended stationary cells the coupling increased with time. The rate of increase in mechanical coupling was similar in the exponentially and overnight culture. As the two results are qualitatively similar this would argue against the existence of preformed connections that were transferred before the experiment starts in washed and re-suspended stationary cells.

Text describing the mechanical coupling in the exponentially grown bacterial suspensions has been added to the Results on p. 6, l. 183-188.

The coupling strength in the exponentially grown cells, however, was higher than expected and larger compared to the washed and re-suspended overnight cells. We have done further experiments to explain this. From our previous experience working with *B. subtilis* we knew that the exponentially grown cells are much more sensitive to environmental perturbations compared to washed and re-suspended stationary cultures. In particular, exponentially grown *B. subtilis* cells may lyse in response to different environmental stresses (Danevčič et al, 2016). To check for cell lysis, shaking of the exponentially grown bacterial suspension was stopped at predefined incubation times, and 2 ml of the bacterial suspension was put to rest in cuvette at room temperature. Optical density of the incubated samples was measured at regular intervals. The absence of shaking reduced oxygen diffusion to the bacterial culture and cells particularly at high biomass density experienced oxygen deprivation, which in the case of aerobic *B. subtilis* cells can induce a severe stress. As given in **Fig. 3a** cells at high biomass density lysed. Although initially bacterial cells at higher biomass densities continued to grow, they soon started to lyse. On the other hand, at low cell densities, similar to the optical densities in the optical tweezers experiments, we did not observe cell lysis. It is important to note that cells in optical tweezers experiments were kept prior to the measurements at 4°C. Under cold conditions the cell lysis was not pronounced even at high cell densities for a prolonged period of time. If exponential cells were washed and re-suspended in PBS buffer, cells started to lyse (**Fig. 3b**). This was, however, different to the stationary cells that were washed and re-suspended in PBS, where cell lysis was much less pronounced. When stationary cells were incubated in PBS for 2.5 h, the mechanical coupling did not change significantly (**Supplementary Fig. 12**). At the end of the incubation in PBS the coupling was 30 μm , which was lower compared to cells incubated in the growth medium (**Fig. 2b**). This implies that increased coupling measured in the growth medium was due to the new production of the extracellular matrix material.

The text describing this has been added to the Results on p. 6-7, l. 188-202. The cell lysis has been described in the Materials and methods on p. 19, l. 676-692.

The indication that the exponential cells are more prone to cell lysis compared to the stationary cells was further checked with TOTO-1 nucleic stain. Stained exponentially grown samples were not incubated for 15 min as recommended by the manufacturer, but were immediately taken for observation under the microscope. Two minutes after sample preparation approximately 2 % of the exponentially cells were intensively fluorescing indicating that cell membranes were compromised. This is approximately an order of magnitude higher than in the overnight bacterial suspension. With increasing time of microscopy more cells start to fluoresce (**Supplementary Fig. 13**). After 30 min of microscopic observations small fluorescent corpuscular bodies appeared in the vicinity of the dying bacterial cells. Small corpuscles eventually formed a halo of swarming fluorescence bodies around a cell. Most of the fluorescence bodies were tethered to the dying cell. A fraction of fluorescence corpuscular bodies moved freely in the medium. We have observed that dying cells were frequently connected with long fluorescence filaments not present at the beginning. Using SEM microscopy (**Supplementary Fig. 14**) one could observe a progressive morphological decay. These results explain why live/dead test regularly fails on exponentially grown *B. subtilis* cells, but give meaningful results in the stationary phase. We have repeatedly observed that the vast majority of the exponentially grown cells turn red in live/dead assay. The results indicate that *B. subtilis* cell membranes may become compromised during the relatively short incubation period which is required according to the manufactures protocol for live/dead assay. Shaken exponential cells, on the other hand, continue to grow and reach the stationary phase, when cells are less sensitive to environmental perturbations. This explains why the majority of stationary cells are green with intact membrane after live/dead assay performed in the stationary phase.

The new findings are described the Results on p. 7, l. 204-218 and are discussed on p. 11-12, l. 345-354.

Given the fact that the exponential cells are more susceptible for cell lysis it is possible that the released cell material contributes to the mechanical coupling and consequently to the mechanical coupling of bacterial pairs. To demonstrate this exponentially grown cells were lysed. The effective coupling distance increased significantly from $(25 \pm 3) \mu\text{m}$ at the beginning to $(50 \pm 6) \mu\text{m}$ after 60 min of cell lysis (**Fig. 3c**). This is a strong indication that lysed cell material contributes to the mechanical coupling via the extracellular matrix.

The effect of cell lysis on the mechanical coupling is described in the Results section on p.7, l. 220-226.

The effect of cell lysis could be present also in other bacterial suspensions. To check for that we have measured the optical density in other unshaken bacterial cultures as well. The cell lysis of exponentially grown cells was less pronounced or absent in other

bacterial species (**Supplementary Fig. 15**). The results indicate that optical density increased in *E. coli*, *V. ruber*, and *P. aeruginosa*, did not change in *P. fluorescens* or *P. stutzeri*, and slightly decreased in *S. aureus*. It is important to note that the mechanical coupling was present in bacterial suspensions also in the absence of massive cell lysis (**Movies 2-7**). The coupling was not necessarily weaker. For example, a rather strong coupling was observed in *P. aeruginosa* bacterial suspensions prior to visible aggregate formation.

The new text describing the lysis results is given on p. 8, l. 228 - 235.

When sampling the cultures the authors store the samples at 4 °C until evaluation by optical tweezers. This is problematic as most bacteria will aggregate and precipitate when chilled in static conditions. This will add yet another possible false positive to the setup.

Samples for optical tweezers experiments were stored at 4 °C to prevent cell lysis. All the measurements on optical tweezers have been done at room temperature. Storing the samples at 4 °C will inevitably decrease the metabolic and swimming activity. As determined by DIC and fluorescence microscopy, cells did not aggregate or lysed at low temperatures. We have compared the mechanical coupling of the samples that have been at room temperature with samples that have been stored at low temperatures prior to the measurement. In both cases cells were mechanically coupled. The samples that have been at room temperature had slightly higher but not significant mechanical coupling than samples that were kept at low temperature for the same duration. The results demonstrate that storing cells at low temperatures did not induce the viscoelastic effect observed.

We have enhanced the text in Materials and Methods to make this point clear on p. 15, l. 507-512.

I miss control experiments of the influence on aggregation and coupling over time, by the addition of both Sodium azide and NaN₃ (figure 1e. only shows the effect of Sodium azide at 0 h).

We agree that this could be improved. We have done further experiments with the addition of sodium azide (NaN₃) at different times during the growth experiment. Sodium azide was added at the beginning of the experiment at t₀. Cells were shaken for 2.5 hours prior to the measurements. The results indicate the coupling did not change significantly during the incubation after the addition of sodium azide (**Supplementary Fig. 9**). Next, Sodium azide was added to the culture which was growing for 1.0 h. The mechanical coupling was higher as expected but did not changed significantly after incubation in sodium azide. Similarly, if sodium azide was added after 2.0 h of incubation there was no significant change of mechanical coupling after the incubation in sodium azide. We did not observed cell aggregation after sodium azide addition.

The new text describing these experiments have been added to Materials and methods on p. 13, l. 412-419, as well as in the Results on p. 5, l. 156-159.

The section on enzymes puzzle me. What are those enzymes the authors want to test? If the authors want to test the effect of spend media on the cultures, why don't they sterile filter the media? The effect of the spend media could be linked to a variety molecules into the medium; fatty acid, QS molecules, and other metabolic products.

Spent medium may contain bacteria, a variety of macromolecules (i.e. nucleases, glucanases, lipases, proteases), small organic molecules, and salts. We have not chemically analyzed the composition of spent medium. Both sterile filtration and centrifugation remove bacterial cells from spent medium, albeit with different efficiencies, but retain macromolecules and smaller molecules that may alter the effective mechanical coupling.

We have rewritten the text in order to make this point clearer in the Materials and methods on p. 17, l. 576-589.

Results:

*I am missing the results *P.stutzeri*, *V. ruber*, *E.coli*. at least in Supplementary Materials.*

We have extended the list and included also *Pseudomonas aeruginosa* and *Staphylococcus aureus*. We now show movies of all tested bacteria in **Supplementary Movies (1-7)**.

Figure 2. It is very hard to identify flagella in the mist of dehydrated exo-polysaccharide. Are the authors positive that what they mark as flagella are these? In the top left panel, there would in this case be very large amounts of flagella. Flagella anti-bodies coupled with gold particles would be improve the detection.

A new **Fig. 1** where flagella are clearly visible at t_0 has been made.

Discussion:

Line 199 There are in fact quite a lot of studies published on non-attached aggregated biofilm in liquids.

Indeed. Though several bacteria have been described that form visible aggregates in liquid cultures (i.e. *Pseudomonas aeruginosa* (Schleheck et al. 2009, Déziel, et al, 2001), *Sinorhizobium meliloti* (Sorroche et al., 2010), *Micrococcus luteus* (Voloshin and Kaprelyants, 2005), *Campylobacter jejuni* (Joshua et al., 2006), *Streptococcus pyogenes* (Frick et al., 2000), *Staphylococcus aureus* (Haaber et al., 2012)) the existence of

mechanically coupled seemingly individual cells in dilute suspensions has not been convincingly demonstrated so far.

We remedy for this in the Discussion on p. 9, 268-273.

Line 214-216 This sentence really summarizes the biggest problem with the study, the authors are studying the formation of connections, but are studying whatever is transferred from the ON.

As discussed above we have done quite some thinking and new experiments in this direction. This is important to our manuscript and microbiology in general as transfer of microbial cultures by inoculation is one of the pivotal tenets in microbiology experimentation. With the new control experiments we have shown that diluting bacterial overnight cultures in the new growth medium by inoculation does more than just transferring cells to the new environment. 100 fold dilution decreased the mechanical coupling between bacterial pairs in the new environment but the coupling nevertheless remains very large (50 ± 5) μm . A significant part of the coupling could be removed if cells were washed and re-suspended in the medium (18 ± 2) μm . The mechanical coupling grows over time both in the truly exponential and washed and re-suspended stationary cells suggesting that new material is constantly added to the extracellular matrix which will eventually mature reaching the new stationary phase. We don't think that transferring bacteria with their local environment during the inoculation is by default a drawback.

The new text has been added to the discussion on p. 9, l. 281-288.

Line 217-219. There are no basis for the statement that growth rate should be affected in new cultures based on the current study, and no reference which can substantiate this are provided.

We have done some new control experiments which indicate that washed cultures have longer lag phase compared to simply diluted bacterial cultures. However, we felt that this would need more experimental effort to prove and therefore we have removed this statement from the manuscript.

The authors hypothesize that the formation of mechanically coupling may decrease the effect of antibiotics do to extra cellular crowding at high cell densities. This would be such a simple experiment, why was it not performed? The hypothesis of mechanically coupling may decrease the effect of antibiotics does not take any metabolic factors into account. It is well know that metabolic activity highly influence the affectivity of most types of antibiotics. In cultures with increased cell density, metabolic activity is most likely to decrease rapidly as well. I do not find any base in this study that underpins this direct coupling could explain an increased antibiotic tolerance, which cannot also be explained by shifts in metabolic activity.

We agree that this is a challenging hypothesis that would need more thorough experimental support. As things stand now, it is likely that above and below the percolation threshold the amount of the extracellular material that mechanically couples cells is different. This may have a significant effect on effective antibiotic concentration in bacterial suspension. However, given a large uncertainty in the metabolic state of bacteria it is indeed premature at this stage to speculate about the effect on antibiotic efficiency and we have removed this part from the new manuscript and only mention it as an afterthought in the discussion on p. 12, l. 366-367. Nevertheless, it would be worthwhile to put more effort in the future experiments to see if this adds in the interpretation of antibiotic effect at low and high density cultures.

Minor points:

Line 38 Please support statements with reference

We have added a new reference Aguiar et al., 2011.

Line 89 It is called Vibrio ruber.

Yes. We have corrected for spelling errors in the new text.

Line 139-142 Please support statements with reference

This part of the manuscript has been rewritten. During the experiment bacteria divided a couple of times (**Supplementary Fig. 10**), which indicates that the mechanical coupling develops early in the growth phase. This is in sharp contrast to the currently accepted view that viscoelastic behavior typical for interconnected bacterial community only occurs at a transition into a stationary phase (López et al., 2015, Portela et al., 2013, Patrício et al., 2014).

The new text is given on p. 5, l. 160-164.

Line 212-214 Please support statements with reference

The percolation model has been, based on the comments by Reviewer N°2, removed from the manuscript and this does not apply any longer.

Line 229-231 revise the structure of sentence

Due to the insufficient experimental support for the quorum sensing part we have decided, similar to your comments on the antibiotic effect, to remove this part from the text and mention it as an afterthought and plan for future work. In the revised text we have instead put more effort to describe the new phenomenon and give more experimental support for the observed effects.

Line 292 LB are not albeit for Luria-Bertani, but Lysogeny broth.

We have changed the text.

Line 292 Why are all strains grown at 28°C? E.coli optimal are 37°C, Bacillus subtilis are reported to be in the range 34-37°C. Vibrio ruber at 40°C.

Not all bacterial strains were grown at 28 °C. For instance, *E. coli* and *S. aureus* were grown at 37 °C. The mechanical coupling for *B. subtilis* grown at 28 and 37 °C has also been checked and no significant difference in coupling strength has been observed.

Line 308-310 The authors fails to explain the rationale behind the addition of sodium azide. Why blocking the respiration?

We have used sodium azide to block respiration and consequently stop biosynthesis of new extracellular material. Another reason to asphyxiate bacteria was to stop their swimming motion which tends to disturb optical tweezers experiments as optically trapped bacteria were able to escape from the trap.

We have corrected for this in the new text in Materials and methods on p. 13, l. 412-419.

Line 468 Temp. written in parentheses.

Although in the literature one frequently encounters notation in the form of $x \pm SD$ °C, it is in fact incorrect. Such notation implies that only SD has a unit in °C. When written in parenthesis such as $(x \pm SD)$ °C, the meaning is clear $(x \pm SD)$ °C = x °C \pm SD °C. This can be checked on <http://physics.nist.gov/cuu/Uncertainty/examples.html>.

Figure 1c r²-value of fitted model would be beneficial.

We have added r^2 to the Figure caption of new **Supplementary Fig. 3**.

Reviewer #2

The article by Sretenovic et al. is concerned with the emergence of extracellular matrix coupling mechanically bacteria during the early growth of a planktonic population. Using TEM micrograph, they witness extracellular material attached to the flagella and the cellular body, even the early phase of the growth. The central claim of the paper is that they are able to measure a mechanical response due to this very tiny network which can significantly couple bacteria over large distances. From these measurements, they extract an interaction distance that they show to be growing with time as the bacteria concentration increases. They relate the growth of an interaction length with the existence of intercellular communication and quorum sensing. They claim that when the culture has reached a so-called “percolation threshold” that they define through a model, the communication is switched on. The author’s final claim is that this mechanistic explanation can solve a long standing problem of sudden resistance of bacterial populations to antibiotics.

This is a - priori an interesting paper, from the biological point of view the questions are relevant and well explained in the text. I agree with the authors on the crucial importance of the result if their claim is true. The author present a lot of data and follow a clear reasoning essentially based on the existence of a coupling mechanical length appearing during the early phase of the growth process. They present a model to correlate this length to the emergence of extra-cellular matrix connecting mechanically the bacteria. However due to the importance of the problem and also the high level of the journal they want to publish in, it also requires some high level of exigency on the significance of the measurements. There are many points that I do not understand, some on which I am skeptical and even some points I completely disagree with. I would be extremely pleased if the authors could reply precisely and convincingly to these remarks. At the present state, I do not think this paper is publishable in Nature Communications.

Let us review the different elements of proof.

1) The material existence of an extracellular matrix.

The micrographs of Fig. 2 seem indeed to show the existence of extra cellular matrix growing with time and especially at early time (2.5 h) one can observe some deposit. For me this is a central point which seems irrefutable provided no artifact is coming from the method. I do not see why it is not coming earlier in the exposition of arguments. I am not sure of the literature of the subject but is it something original from this paper to witness such extracellular matrix at such early time of growth? From the micrograph is it possible to estimate the volume content of this extracellular material? A related question is on what we see. Is the image presented statistically characteristic of the status at any point on the sample, or did the author have chosen some particular place especially eloquent to push their point?

As suggested by the reviewer, we have rewritten the text and presented this earlier in the exposition of the arguments. To further characterize the extracellular material we have used SEM microscopy (see **Supplementary Fig. 1**), which indicates that extracellular

material of *B. subtilis* may attach to the cell forming a smeared halo of material around the cell. The fraction of cells with smeared halo around the individual cells increases with growth of bacteria in suspension. Although micrographs indicate a connection of the extracellular material to the cell surface, we cannot rule out, due to the nature of sample preparation, that cells act as nucleation centers facilitating the assembly of the extracellular material around the cell. We are not sure if the estimate of the volume of the extracellular halo can be reliably determined. This is a hydrocolloid like material that may shrink during the preparation of the sample. The extracellular matrix was present in other cells as well. For instance, in the exponentially grown *E. coli* cells there was less extracellular matrix attached to cells with different electron density that did not resembled the electron density of cell constituents (**Supplementary Fig. 2**).

We agree with the reviewer that showing micrographs without a proper statistics is a convenient way to push the point. To avoid this we have taken pictures at different magnifications to have a broader perspective about the distribution of the extracellular material, but we did not show them in the submitted version. We attach a low magnification figure to this point by point replay to present statistical character of the selected micrographs in **Fig. 1**. At low cell density the extracellular material was unevenly distributed. With higher cell density and higher amount of extracellular material the distribution became more evenly spread out. At the end of the exponential growth phase the micrographs were covered with the extracellular material.

Low magnifications of TEM micrographs of *B. subtilis* wt cultures after 0, 2.5, 5, and 8 h of incubation. Several viewing screens are given for each time point to give a better sense of statistical nature of extracellular material distribution. A single viewing screen at a given time point has been selected to make a high magnification representation of extracellular matrix material which was presented in Fig. 1. Inevitably, in zooming in one has to make a selection what is representative of the sample at large. We believe that the essential feature of the growing extracellular matrix network has been captured in **Fig. 1**.

2) *Collective hydrodynamics dismissed or not?*

*The first thing to dismiss is the possibility of hydrodynamic coupling associated with the early existence of collective swimming of the bacteria that provides non-standard hydrodynamic coupling eventually reinforcing the response to a local perturbation and being an effect very different from the present interpretation. The authors quote the work of Gachelin et al (ref.17) on rheology. Note that in the same group there was recently measurements that may be pertinent to discuss along with these results. Lopez et al. (Lopez et al Phys. Rev. Lett. **115**, 028301 (2015)) show that classical rheology can be done at much lower shear rate (0.1 s^{-1}) that what is presented here and indeed these authors have shown a very peculiar rheological response for a sheared planktonic suspension of *E.coli* for about the same density range. Note that in their case, they use a fresh motility buffer to re-suspend the bacteria which in principle is cleaned from any extracellular matrix.*

López et al., 2015 measured the rheological response of active cell suspensions of *E. coli* at low shear rates i.e. 0.01 s^{-1} . However, they used bacterial cell densities $> 1.1 \times 10^9$ cells/ml. This is much higher (approximately 2 orders of magnitude) than in our case where we have done measurements at bacterial densities in the range between 1.0×10^7 and 6×10^7 cells/ml. Lower cell density used explains why we were not able to detect a meaningful viscoelastic response by classical rheology.

We are aware of the collective swimming behavior of the bacteria, in particular of the *B. subtilis* cells at high cell densities (in excess of 10^9 cells/ml) upon transition from random swimming to transient jet and vortex patterns in the bacteria/fluid mixture (Wolgemuth C.W. 2008). The observed bacterial turbulence generated by random swimming patterns decreases viscosity of the medium suggesting high Re fluid behavior at high bacterial densities. Therefore change to the inertial flow behavior should be more pronounced with increasing cell concentration, which is opposite to what we have observed. In our experiments, performed at much lower cell densities, the strength of the coupling increased with time of incubation. In addition, we have not observed collective swimming behavior at low cell densities. Therefore we think that at low cell densities collective swimming motion is not contributing to the observed coupling effect.

We discuss this in the new manuscript on p. 10, l. 291-296 and have included the reference to López et al. 2015 work.

a) First the statement that classical rheology method cannot access the suspension's rheology is not exactly true as shown by Lopez et al.

We have measured the rheology of diluted bacterial suspensions with rotational rheometer, but were not able to detect significant effect of bacterial growth on viscoelasticity. If we have concentrated the early exponential cells by 10 fold with centrifugation and then re-suspended pellet in the spent medium a pseudoplastic flow behavior was observed. The amplitude oscillatory tests on such concentrated bacterial

suspension indicated a weak elastic modulus. We do not show these results in the new manuscript as we were able to measure *in situ* viscoelastic response of the extracellular matrix material by optical tweezers which provides a better indication of viscoelastic effect as will be explained below.

b) Second, I think that the measurements presented in fig. 1.a, are useless essentially because they are not done in the right shear rate regime (around 100 s⁻¹ !). At this shear rate basically any fine interconnected structure is likely to be destroyed and more importantly, the method is not adapted as to be able to probe viscoelastic response. In this perspective one should do oscillatory or time step response. I will come back to that but I do not see the interest in the main part of the text of such a rheological measurement obviously nonadapted to the question. Sretenovic et al. use a PBS solution which in principle should get rid of extracellular matrix as in Lopez et al. and see a much weaker if any, long range hydrodynamic coupling.

At the shear rate of 100 s⁻¹ it is likely that the delicate extracellular structure will be destroyed. However, the flow curves were measured by starting at low shear rate (i.e. 10⁻² s⁻¹) which allows one to probe the delicate structures if present and then progressed to higher shear rates. The viscosity of the bacterial suspension was only slightly larger than viscosity of the medium and we were not able to obtain reliable data below the shear rate of 1⁻¹s. We agree that the flow curves do not add much information to the manuscript and have been removed.

c) What is the guaranty that the bacteria are still active in this PBS medium? Is there some possibility to characterize the swimming activity by assessing the mean swimming velocity when these measurements are done? Along those line, Gachelin et al (New Journal of Physics, 16, 025003 (2014) have measured a correlation length coming from collective swimming of the bacteria and which grows linearly with the bacteria volume fraction. Therefore, collective swimming effect inducing hydrodynamic coupling may be important as well, along with the growth of an extracellular matrix.

Stationary cells can survive a prolonged suspension in PBS buffer or saline suspension (Liap and Shollenberger, 2003). However, as mechanical coupling can be affected even if a fraction of cells lyse we have done further experiments to see how cell survive in PBS. Both overnight cells and exponentially grown cells were re-suspended in PBS buffer and put to rest in the cuvette. The results are given in **Fig. 3b**. If exponentially grown cells were re-suspended in PBS, optical density started to decrease immediately after re-suspension in PBS buffer indicating cell lysis. Cells progressively lost swimming velocity and were not mobile. Cell lysis increased the mechanical coupling (**Fig. 3c**). On the other hand, when the overnight cells were re-suspended in PBS the optical density changed much less over 2 h of incubation. Consistently, the mechanical coupling was significantly lower than in the exponentially grown cells. As the majority of cells were not actively moving the effect of swimming inducing the hydrodynamic coupling was not very important in PBS environment. When stationary cells were incubated in PBS for 2.5 h the

mechanical coupling did not change significantly (**Supplementary Fig. 12**). At the end of the incubation in PBS the coupling was 30 μm , which was lower compared to cells incubated in the growth medium (**Fig. 2b**). This implies that increased coupling measured in the growth medium was due to the new production of the extracellular matrix material.

The effect of PBS and cell lysis is described in Results on p. 6-7, l. 195-202.

d) Is there a possibility to “kill” or asphyxiate the bacteria in the middle of the growth phase (and not at the beginning as in Fig.1e) to try to separate collective swimming hydrodynamics from polymeric matrix effects?

Depending on the type of metabolism and bacterial species it is possible to asphyxiate the bacteria. In case of aerobic respiration metabolism present in *B. subtilis* it is possible to stop active metabolism by addition of sodium azide. We have done a series of experiments when sodium azide was added at different times during growth experiments. Cells stopped moving after the addition of azide. The new results are presented in **Supplementary Fig. 9**. Sodium azide was added at the beginning of the experiment at t_0 . Cells were shaken for 2.5 hours prior to the measurements. The results indicate the coupling did not change significantly during the incubation after the addition of sodium azide. Next, sodium azide was added to the culture which was growing for 1.0 h. The mechanical coupling was higher as expected but did not change significantly after incubation in sodium azide. Similarly, if sodium azide was added after 2.0 h of incubation there was no significant change of mechanical coupling after incubation in sodium azide. We did not observe cell aggregation after sodium azide addition. The results suggest that the effect of collective movement on mechanically coupled cell is not very large at low cell densities in bacterial suspensions.

The new text describing these experiments have been added to Materials and methods on p. 13, l. 412-419, as well as in the Results section on p. 5, l. 156-159.

However, it proved to be more difficult to asphyxiate *E. coli*, *V. ruber* and *P. aeruginosa* cells with sodium azide. After the addition of sodium azide cells remain active for quite a while. In those bacteria which have alternative metabolic strategies repressing oxygen respiration is not enough to stop their metabolism and swimming.

3) Measurement method to extract a coupling length

I agree with the authors that the local rheology measurement are a very useful tool to assess the mechanical coupling in the suspension. As I already discussed, oscillatory response is indeed a good way to probe viscoelastic coupling. I am convinced from the data presented by the authors that something indeed is going on when the concentration is increased. My main concern is on the methodology and on the analysis of the data allowing to extract a meaningful “typical length” representing a mechanical coupling. This is a central element that determines non-only the pertinence of this article and also

the subsequent discussions in the manuscript based on the physical significance of this length.

With the new experiments performed during the revision we are reassured that the mechanical coupling is a widespread long-range phenomenon that exists in different bacterial cultures, cell densities, growth conditions and methods of cell suspension preparation. The fragile nature of intracellular material, however, remains difficult to analyze. We have used coupling length as a parameter to characterize the intensity of the coupling between a pair of isolated bacteria. However, the sensitivity of the experimental setup sets a threshold on the maximum length over which the coupling can be detected. The extent of the coupling does not stop at typical length and can go beyond the coupling distance as defined in the manuscript. The determined distance between pairs of optically trapped bacteria is therefore not an absolute distance but an effective coupling distance as will be discussed below.

In the revised text we define the coupling distance as the effective coupling distance as defined in the Materials and methods on p. 15, l. 499-501.

From the oscillatory response, what the authors essentially assess is the intensity of the coupling (displacement) with distance. I have two major critics to do.

a) From what I see for example in fig 1c, there is no strong argument in favor of a typical coupling length at least the way it is extracted from a linear fit. To illustrate this essential rebuttal, I have plotted the passive bacterium amplitude response R multiplied by the interbacteria distance r as a function of the inter-bacteria distance r . From what is seen in the subsequent figure, a response as $R \propto 1 / r$, meaning essentially an effective hydrodynamic coupling in form of a power law decay (and no typical length scale) , is not at all impossible.

Data $R*r$ versus r extracted from fig. 1c.

With increasing distance r between pairs of optically trapped bacteria a point is reached when the amplitude of the passive bacterium cannot be reliably determined. However, this does not imply that the coupling stops there. It is likely that the coupling extends

beyond this distance. So coupling distance as defined in the text could indeed be longer and there may be no typical length scale present. We have, as suggested, increased the distance between the bacterial pairs up to 40 μm when the response of the passive bacterium was 0.11 μm (**Supplementary Fig. 3**). This is probably as far as one can reliably go. With the new data we have plot the A^*r versus r . The data are presented in (**Supplementary Fig. 18**). The results do not support a conclusion that $A \propto 1/r$.

We have added this to new manuscript on p. 9, l. 259-264.

The hydrodynamic properties of all, but the simplest colloidal systems, however, are controversial and have been a subject of considerable debate (Segre et al., 1997). A key factor in this uncertainty has been the intrinsically long-ranged nature of the hydrodynamic coupling between solid particles. It has been demonstrated for a pair of individual polymer-coated polymethylmethacrylate particles ($d = 1.3 \mu\text{m}$) using a passive microrheology that hydrodynamic coupling may extend up to 20 μm (Paul Bartlett et al., 2001). Bacterium with its numerous extensions (i.e. lipopolysaccharides, lipoproteins, pili, fimbria and flagella) is effectively a larger object than the cell body itself. Therefore it is not impossible that *B. subtilis* cells feel each other also hydrodynamically at distances of around 15 to 20 μm , which in most cases was the initial coupling distance.

We discuss this on p. 10, l. 304-314.

b) The second question is why the authors do not assess the phase response and do not vary the oscillation frequency to really test a viscoelastic signature? This is indeed a standard way to prove a visco-elastic coupling. Note that in this case, there is no clear length scale coming out from such a model.

I think that if the authors really want to push forwards convincingly the existence of a coupling distance (for whatever it means) they have to push the measurements over larger spanning distances so that one can really separate a response producing a convincing length scale from any power law decay. If the authors want to push forward the existence of an effective viscoelastic coupling growing with concentration, my impression is that they have to do it is the standard way and extract from the frequency response the viscous and elastic parameters.

Assessing phase response in low viscoelastic liquids is not easy and has to the best of our knowledge not been attempted *in situ* in dilute bacterial suspensions. We have measured the phase response in low density bacterial suspensions, as suggested. Active one-particle microrheology measurements were performed using a precise computer-controlled sinusoidal modulation of trapping-beam position and synchronous measurement of particle position. This makes it possible to determine storage and loss modulus, G' and G'' , respectively. The results (**Fig. 4**) indicate that the extracellular matrix material is viscoelastic in nature. The loss modulus was not significantly different from the medium. There was, however, a significant elastic contribution of the extracellular matrix that was not present in the growth medium. Both storage and loss modulus increased with

increasing frequency. The frequency response observed was typical for the behavior of weak viscoelastic liquid composed of unlinked polymers (Mezger, 2011).

The experiment is described in Material and methods on p. 15-16, l. 514-525, the results are presented on p. 8, l. 237-248 and are discussed on p. 10-11, l. 319-327.

4) Percolation model

Based on the assessment of a coupling length, the authors put forwards a percolation model to interpret their biological data and try to prove an onset of communication between the bacteria once the percolation threshold is reached. Such a relation between a topological transition and a global onset of communication inducing a quorum sensing response would indeed be very important and novel conceptually. I think this is the very weak point of the paper. Even with my previous doubts on the real existence of a length, the static percolation picture they present is hard to buy with swimming bacteria that would carry along their path this “corona” of extracellular matrix. I do not see in the data any convincing element proving the validity of such a picture. Moreover, I do not see any clear and convincing relation between a percolation threshold, even defined as the authors do, and the emergence of a quorum sensing signal.

The SEM micrographs (**Supplementary Fig. 1**) do not exclude entirely the percolation model. Cells may act as nuclei for condensation of extracellular material forming a corona of extracellular matrix. The material may start to overlap at higher cell densities as the percolation model predicts. Similarly, we have shown with TOTO-1 nucleic stain that extracellular material, which is released from cells during cell lysis, may form a halo around the cells (**Supplementary Fig. 11, 13**). The fluorescent filaments that emerge from the dying cells organize this halo into an extracellular network. Due to the fragile nature of the viscoelastic material the actively moving bacteria may easily plough through such an extracellular matrix. It is only in latter phases of biofilm development that the viscoelastic matrix is reinforced and condenses which prevents active motion of bacteria. However, given the uncertainty in the determination of the typical length as discussed above the simple percolation model as described in the submitted text is an oversimplification and we have removed it from the new manuscript.

The demonstration of a relation between a percolation threshold and the emergence of quorum sensing response is complex and unsolved question. We agree that at the moment the experimental support for the involvement of mechanical coupling in quorum sensing is weak. To make convincing connections a different set of quorum sensing reporters would be needed. We do not have such reporters ready to use in lab. As the main objective of the paper is a demonstration of viscoelastic network connecting neighboring cells in the dilute bacterial suspensions, we believe that this part of the manuscript can be left out and be only mentioned as an afterthought and stimuli for new experiments in the field.

We present this at the end of Discussion on p. 12, l. 363-369.

Reviewer #3 (Remarks to the Author):

This study reports the observation of mechanical coupling between bacteria at nominally rather low densities due to extracellular substances. This is a very interesting observation that could indicate that our thinking about dilute bacterial suspensions is mostly wrong and could thus potentially be highly influential. However, extraordinary claims need extraordinary evidence and I am not sure that the required threshold of evidence is reached here; the importance of the results could also be much smaller (merely indicating that experiments must be done more carefully than what is usually done). In my opinion, the authors show interesting data in support of an exciting hypothesis, but not up to the point where they convinced me. Partly this is due to how the paper is presented (the small number of control experiments is shown very late), partly this is due to the lack of control experiments. In my opinion, the paper could be interesting enough to be considered for publication, but substantial additional work would be needed.

*1) The first qualitative experiments (suppl. movies) did not convince me. Why is a different bacterium (*Pseudomonas*) used here than in the rest of the paper? The authors claim that they did this experiment with different species, but do not show any data.*

We now show movies of all tested bacteria. In our opinion, the recorded movies clearly demonstrate the viscoelastic nature of the extracellular matrix and are strong and the most unbiased evidence that convince the potential reader to the existence of the invisible network in different planktonic bacterial cultures. In addition, we have extended the list of bacteria that have been tested with some of the standard medically relevant model organisms such as *Staphylococcus aureus* and *Pseudomonas aeruginosa*. Furthermore, we have used a new methodology of data analysis in the revised manuscript using a single particle active microrheology to determine the non-linear response of bacterial clusters. Instead of using a pair of bacteria we have determined the extent of viscoelastic mechanical coupling on a cluster of bacteria. The reaches of coupling distances have been estimated in different bacterial suspensions and range from 60 to 140 μm . The results suggest that coupling in different bacteria may extend to very large distances.

The new results are given on p. 3, l. 87-103.

Moreover, the authors claim to see a weak viscoelasticity of their suspensions when using optical tweezers, but not with other methods. They explain this by the higher sensitivity of optical tweezers, but to any non-expert in optical tweezers, an alternative explanation would be that this is an artefact of the method. I think any control experiment that shows a dependence on the medium (as the authors do later, e.g. replacing the medium with PBS or digestion of extracellular substance) would help a lot here, as it would show that the effect is specific to the bacterial cultures and not argue against artefacts. Ideally this would be accompanied by EM images of the suspensions.

Several new experiments have been done to address these issues. We have done frequency tests with optical tweezers to obtain *in situ* viscoelasticity of the extracellular matrix (**Fig. 4**). The results for the first time show microrheology of local bacterial environment in dilute bacterial suspensions. We have concentrated diluted bacterial suspensions 10 fold with centrifugation and then re-suspended pellet in the spent medium and observed a pseudoplastic flow behavior indicative of viscoelastic behaviour by classical rotational rheometer as well. The amplitude oscillatory tests on such concentrated bacterial suspension indicated a weak elastic modulus. Since centrifugation may have changed the flow behavior of the extracellular material we do not show these results. We have done further tests of mechanical coupling on the truly exponential cells. We have replaced the medium with PBS and show results for the exponential and stationary cells, which indicate the complexity of the coupling phenomenon (**Fig. 3b**, **Supplementary Fig. 12**). We have made new SEM and TEM images and show the attachment of extracellular matrix material on cell surface. We have shown that extracellular matrix grows with time of the incubation for other bacteria as well (**Supplementary Fig. 2**). We have evaluated the effect of cell lysis on the mechanical coupling effect. The new experimental evidence would argue against artifacts.

In general, I think the authors need to convince the reader first that the effect they see is real. The present version of the manuscript is written from the standpoint that this effect exists and can be further studied. I think a change of standpoint towards establishing the existence of the effect and convincing the reader would be helpful. The authors say on p.3: "The results imply that different bacterial species are able to form cohesive viscoelastic networks in the dilute monoculture bacterial suspensions with persistence length of up to 40 μm ." At this point, this is a speculation, and cannot be concluded yet.

We have changed the standpoint in the new manuscript to more clearly establish the existence of the viscoelastic effect in dilute bacterial suspensions, which is indeed the major point of this work. In this respect, we have done several new experiments, added new bacterial strains, which are of general interest, re-analyzed the data, determined *in situ* the storage and loss modulus of the local bacterial extracellular matrix, and increased the number of control experiments.

The upper coupling distance of 40 μm was estimated for a pair of individually optically trapped bacteria. With new control experiments, we have measured even larger coupling distances between pairs of bacteria (i.e. 50 μm in lysed samples **Fig. 3c**). Furthermore, we have applied a single particle active microrheology to determine the non-linear response in the cluster of bacteria. If one, instead of using bacterial pairs, determines coupling in bacterial clusters where several bacteria interconnect the coupling effect is magnified and the estimated coupling length from the movies is even larger (from 60 to 140 μm). The new experiments reinforce the original idea of viscoelastic coupling in dilute bacterial suspensions and open several new questions worth to be further studied as is now discussed on p. 12, l. 361-369.

2) *In the active-passive trapping experiment (Fig. 1), a control would be needed as well. The authors do a control experiment (Bacteria in PBS buffer instead of growth medium), but they present the data in a completely different form, so no comparison is possible. Why not show some traces as in Fig. 1b for correlated motion at different distances in SYN medium and for uncorrelated motion in PBS? Can an upper limit for the amplitude be estimated for the PBS experiment?*

Likewise, show the Fourier transforms for both cases as well, so a direct comparison is possible.

To convince the reader that mechanical coupling in growth medium and PBS are qualitatively different we have made a new **Fig. 2a**. We present traces for both SYM and PBS. The results indicate that at the same distances between the active and passive bacterium pair the bacterial cells are strongly coupled in SYM and only very weakly coupled in PBS. Showing Fourier transforms for the passive bacterium traces in the two media is not meaningful as too few data points were collected when the passive bacterium was in the inactive trap. This is different to the situation in **Supplementary Fig. 16 and 17**, where frequency tests were made to probe hydrodynamic effect and fast Fourier transforms could be obtained.

3) *In general I am wondering whether the bacteria in these experiments are actually in exponential growth phase or rather in lag phase. It seems what is considered here is the first generation (or the first two) after inoculation with some fraction of the extracellular substance responsible for viscoelasticity already present at inoculation. I would suggest a control experiment with cells that are truly in exponential phase due to repeated dilution at low OD with fresh medium. I would expect that the viscoelasticity is much lower then. This could weaken the claim of the authors, but in addition show that one has to be careful how a “dilute suspension” is generated. On the other hand, if the effect is observed in true exponential phase, the claim of the authors would be much stronger.*

This is the central issue on which we have spent most of the time preparing the revised version. Cell washing is used as a standard procedure in most microbiology labs, but as noted has a major drawback that cells are forcefully aggregated during the centrifugation step. As cells compact during the process, it is possible that extracellular matrix is concentrated in the interstitial volume and attaches irreversibly to the cell surface. Although cells were vigorously vortex mixed upon re-suspension, we checked for possible cell-cell aggregation with light microscopy. The aggregates were absent. In addition we performed tests to observe the efficiency of washing to remove the extracellular material. We checked for the presence of eDNA in the stationary cultures that were washed and re-suspended in the growth medium. The presence of eDNA in washed cells was determined with TOTO-1 nanosensitive nucleic acid stain. The results indicate that less than 0.1 % of cells were permeable for the nucleic stain. No fluorescence particles were present in the re-suspended samples and no fluorescence

filaments were attached to the cells (**Supplementary Fig. 11**). In addition, we have prepared washed and re-suspended samples for TEM microscopy (**Fig. 1, Supplementary Fig. 2**). The micrographs indicate regular cells with no indication of aggregated extracellular matrix present in the medium or on the surface of the cells. In contrast, the extracellular matrix of non-washed overnight culture was infested with small fluorescence particles that swarm in the intercellular space (**Supplementary Fig. 11**). Some fluorescence filaments interconnecting fluorescent cells were visible in the samples. Most of the stationary cells were intact and impermeable to the nucleic stain. If stationary cultures were 100 fold diluted, the mechanical coupling between pairs of bacteria remained large (50 ± 5) μm . In sharp contrast, washing stationary cells and re-suspending them in the growth medium reduced the mechanical coupling to (18 ± 2) μm suggesting that washing of cells was efficient.

The text describing eDNA has been added to the Result section on p. 6, l. 167-181. TOTO-1 staining procedure is described on p. 17, l. 598-606.

Although no visible extracellular material was present in washed and re-suspended stationary cells, optical tweezers experiments indicated long-range interconnections (i.e. 18 μm). To further minimize the effect of possible pre-seeded connections, we have as suggested re-grown the inoculum several times to the exponential growth phase (three times to $\text{OD}_{650} = 0.3$) prior to the optical tweezers experiments to obtain truly exponentially grown bacteria. In addition, we have used lower inoculum size (1%) in all subsequent experiments. The results for the mechanical coupling are presented in **Fig. 2b**. The data suggest that mechanical coupling is present also in the truly exponentially grown cells. Similarly to washed and re-suspended stationary cells the coupling increased with time. The rate of increase in mechanical coupling was similar in the exponentially and overnight culture. As the two results are qualitatively similar this would argue against the existence of preformed connections that were transferred before the experiment starts in washed and re-suspended stationary cells.

Text describing the mechanical coupling in the exponentially grown bacterial suspensions has been added to the Results on p. 6, l. 183-188.

The coupling strength in the exponentially grown cells, however, was higher than expected and larger compared to the washed and re-suspended overnight cells. We have done further experiments to explain this. From our previous experience working with *B. subtilis* we knew that the exponentially grown cells are much more sensitive to environmental perturbations compared to washed and re-suspended stationary cultures. In particular, exponentially grown *B. subtilis* cells may lyse in response to different environmental stresses (Danevčič et al, 2016). To check for cell lysis, shaking of the exponentially grown bacterial suspension was stopped at predefined incubation times, and 2 ml of the bacterial suspension was put to rest in cuvette at room temperature. Optical density of the incubated samples was measured at regular intervals. The absence of shaking reduced oxygen diffusion to the bacterial culture and cells particularly at high biomass density experienced oxygen deprivation, which in the case of aerobic *B. subtilis* cells can induce a severe stress. As given in **Fig. 3a** cells at high biomass density lysed.

Although initially bacterial cells at higher biomass densities continued to grow, they soon started to lyse. On the other hand, at low cell densities, similar to the optical densities in the optical tweezers experiments, we did not observe cell lysis. It is important to note that cells in optical tweezers experiments were kept prior to the measurements at 4°C. Under cold conditions the cell lysis was not pronounced even at high cell densities for a prolonged period of time. If exponential cells were washed and re-suspended in PBS buffer, cells started to lyse (**Fig. 3b**). This was, however, different to the stationary cells that were washed and re-suspended in PBS, where cell lysis was much less pronounced. When stationary cells were incubated in PBS for 2.5 h, the mechanical coupling did not change significantly (**Supplementary Fig. 12**). At the end of the incubation in PBS the coupling was 30 µm, which was lower compared to cells incubated in the growth medium (**Fig. 2b**). This implies that increased coupling measured in the growth medium was due to the new production of the extracellular matrix material.

The text describing this has been added to the Results on p. 6-7, l. 188-202. The cell lysis has been described in the Materials and methods on p. 19, l. 676-692.

The indication that the exponential cells are more prone to cell lysis compared to the stationary cells was further checked with TOTO-1 nucleic stain. Stained exponentially grown samples were not incubated for 15 min as recommended by the manufacturer, but were immediately taken for observation under the microscope. Two minutes after sample preparation approximately 2 % of the exponentially cells were intensively fluorescing indicating that cell membranes were compromised. This is approximately an order of magnitude higher than in the overnight bacterial suspension. With increasing time of microscopy more cells start to fluoresce (**Supplementary Fig. 13**). After 30 min of microscopic observations small fluorescent corpuscular bodies appeared in the vicinity of the dying bacterial cells. Small corpuscles eventually formed a halo of swarming fluorescence bodies around a cell. Most of the fluorescence bodies were tethered to the dying cell. A fraction of fluorescence corpuscular bodies moved freely in the medium. We have observed that dying cells were frequently connected with long fluorescence filaments not present at the beginning. Using SEM microscopy (**Supplementary Fig. 14**) one could observe a progressive morphological decay. These results explain why live/dead test regularly fails on exponentially grown *B. subtilis* cells, but give meaningful results in the stationary phase. We have repeatedly observed that the vast majority of the exponentially grown cells turn red in live/dead assay. The results indicate that *B. subtilis* cell membranes may become compromised during the relatively short incubation period which is required according to the manufactures protocol for live/dead assay. Shaken exponential cells, on the other hand, continue to grow and reach the stationary phase, when cells are less sensitive to environmental perturbations. This explains why the majority of stationary cells are green with intact membrane after live/dead assay performed in the stationary phase.

The new findings are described the Results on p. 7, l. 204-218 and are discussed on p. 11-12, l. 345-354.

Given the fact that the exponential cells are more susceptible for cell lysis it is possible that the released cell material contributes to the mechanical coupling and consequently to the mechanical coupling of bacterial pairs. To demonstrate this exponentially grown cells were lysed. The effective coupling distance increased significantly from $(25 \pm 3) \mu\text{m}$ at the beginning to $(50 \pm 6) \mu\text{m}$ after 60 min of cell lysis (**Fig. 3c**). This is a strong indication that lysed cell material contributes to the mechanical coupling via the extracellular matrix.

The effect of cell lysis on the mechanical coupling is described in the Results section on p.7, l. 220-226.

The effect of cell lysis could be present also in other bacterial suspensions. To check for that we have measured the optical density in other unshaken bacterial cultures as well. The cell lysis of exponentially grown cells was less pronounced or absent in other bacterial species (**Supplementary Fig. 15**). The results indicate that optical density increased in *E. coli*, *V. ruber*, and *P. aeruginosa*, did not change in *P. fluorescens* or *P. stutzeri*, and slightly decreased in *S. aureus*. It is important to note that the mechanical coupling was present in bacterial suspensions also in the absence of massive cell lysis (**Movies 2-7**). The coupling was not necessarily weaker. For example, a rather strong coupling was observed in *P. aeruginosa* bacterial suspensions prior to visible aggregate formation.

The new text describing the lysis results is given on p. 8, l. 228 - 235.

For the optical tweezer experiments, cells are stored at 4°C. Does this affect the observations? Is the same coupling seen if cells are directly moved to the tweezers?

Samples for optical tweezers experiments were stored at 4 °C to prevent cell lysis. All the measurements on optical tweezers have been done at room temperature. Storing the samples at 4°C will inevitably decrease the metabolic and swimming activity. As determined by DIC and fluorescence microscopy cells did not aggregate or lysed at low temperatures. We have compared the mechanical coupling of the samples that have been at room temperature with samples that have been stored at low temperatures prior to the measurement. In both cases cells were mechanically coupled. The samples that have been at room temperature had slightly higher but not significant mechanical coupling than samples that were kept at low temperature for the same duration. The results demonstrate that storing cells at low temperatures did not induce the viscoelastic effect observed.

We have enhanced the text in Materials and Methods to make this point clear on p. 15, l. 503-512.

4) In Fig 3, panel d is the most important one in my opinion. This panel contains some crucial control experiments. Specifically, for washed cells and spent medium, supporting

the observation reported here. In addition, it provides the first hints on the genetic basis of the effect showing importance of flagella and lack of importance of eps and tasA.

More than a dozen of new control experiments have been done. We are very pleased to notice that the main conclusions of the work remain unchallenged and that the existence of viscoelastic network in dilute bacterial suspensions has been strongly reaffirmed. However, the new control experiments also reveal some unforeseen effects during the microbial growth. For instance the physiological state of *B. subtilis* bacterial suspension is important. Although qualitatively similar the mechanical couplings of exponential and stationary cells differ. In particular cell lysis is more pronounced in the exponential phase and may release more cell material that contributes to mechanical coupling. The released material contributes to increased coupling (**Fig. 3c**).

We have rewritten the manuscript and put forward the control experiments to strengthen and support the observations.

The discussion of quorum sensing is not very convincing. The authors argue that quorum sensing sets in much later than percolation based on extracellular polysaccharide formation. I would it find more convincing if a quorum sensing reporter had been used that is independent of anything that affects viscoelasticity, but maybe this part of the manuscript can be rephrased with quorum sensing being more of an afterthought on a experiment done for other reasons.

We agree that at the moment the experimental support for the involvement of mechanical coupling in quorum sensing is weak. To make convincing demonstration a different experimental set up with different set of quorum sensing reporters than the ones currently used would be needed. As the main objective of the paper is to convincingly demonstrate the new phenomenon of early viscoelastic network formation in dilute bacterial suspensions we believe that this part of the manuscript can be left out and will only be mentioned as an afterthought and stimuli for new experiments in the field. We have changed text in the new manuscript accordingly.

References

1. Danevčič, T., Vezjak, M. B., Tabor, M., Zorec, M., Stopar, D. Prodigiosin induces autolysins in actively grown *Bacillus subtilis* cells. *Frontiers in microbiology*, **7**, (2016).
2. Schleheck, D. et al. *Pseudomonas aeruginosa* PAO1 preferentially grows as aggregates in liquid batch cultures and disperses upon starvation. *PloS one*, **4**(5), 5513 (2009).
3. Déziel, E., Comeau, Y., Villemur, R. Initiation of biofilm formation by *Pseudomonas aeruginosa* 57RP correlates with emergence of hyperpiliated and highly adherent phenotypic variants deficient in swimming, swarming, and twitching motilities. *Journal of Bacteriology*, **183**(4), 1195-1204 (2001).

4. Sorroche, F. G., Rinaudi, L. V., Zorreguieta, Á., Giordano, W. EPS II-dependent autoaggregation of *Sinorhizobium meliloti* planktonic cells. *Current microbiology*, **61**(5), 465-470 (2010).
5. Voloshin, S. A., Kaprelyants, A. S. Cell aggregation in cultures of *Micrococcus luteus*, studied by dynamic light scattering. *Applied Biochemistry and Microbiology*, **41**(6), 570-573 (2005).
6. Joshua, G. P., Guthrie-Irons, C., Karlyshev, A. V., Wren, B. W. Biofilm formation in *Campylobacter jejuni*. *Microbiology*, **152**(2), 387-396 (2006).
7. Frick, I. M., Mörgelin, M., Björck, L. Virulent aggregates of *Streptococcus pyogenes* are generated by homophilic protein–protein interactions. *Molecular microbiology*, **37**(5), 1232-1247 (2000).
8. Haaber, J., Cohn, M. T., Frees, D., Andersen, T. J., Ingmer, H. Planktonic aggregates of *Staphylococcus aureus* protect against common antibiotics. *PloS one*, **7**(7), 41075 (2012).
9. Aguiar, M., Ashwin, P., Dias, A. Field, M. Dynamics of Coupled Cell Networks: Synchrony, Heteroclinic Cycles and Inflation. *Journal of Nonlinear Science* **21**(2), 271-323 (2011).
10. López, H. M., Gachelin, J., Douarche, C., Auradou, H., Clément, E., Turning bacteria suspensions into superfluids. *Physical review letters*, **115** (2), 028301 (2015).
11. Portela, R. et al. Real-time rheology of actively growing bacteria. *Physical Review E* **87**, 030701 (2013).
12. Patrício, P. et al. Living bacteria rheology: Population growth, aggregation patterns, and collective behavior under different shear flows. *Physical Review E* **90**, 022720 (2014).
13. Wolgemuth, C.W. Collective swimming and the dynamics of bacterial turbulence. *Biophysical journal*, **95**(4), 1564-1574 (2008).
14. Liap and Shollenberger, *Lett Appl Microbiol*, **37**(1), 45-50 (2003).
15. Bartlett, P., Henderson, S. I., Mitchell, S. J. Measurement of the hydrodynamic forces between two polymer–coated spheres. *Philosophical Transactions of the Royal Society of London A: Mathematical, Physical and Engineering Sciences*, **359**(1782), 883-895 (2001).
16. Mezger, T. G. *The Rheology Handbook*. (European Coatings Tech Files, ISBN 978-3-86630-864-0, ed. 3, 2011).

REVIEWERS' COMMENTS:

Reviewer #1 (Remarks to the Author):

I recognize the immense effort the authors gone through to bring the manuscript up to quality in response to my comments and recommendations. The work has elevated the quality of the manuscript vastly. I especially want to recognize the additional experiments performed with several additional species and the changes to the method.

The manuscript and data are at a very high quality

Reviewer #3 (Remarks to the Author):

In the revised version of the manuscript, the authors have added a number of control experiments, specifically a comparison of different growth protocols (incl. repeated dilution) and standard microrheology measurements of viscoelasticity. They have also removed some claims that were too strong based on their experimental evidence.

In my opinion, the paper now demonstrates clearly that the dilute bacterial suspension is viscoelastic and that mechanical coupling between cells (as defined by the optical tweezers experiment) exists. The molecular nature of this coupling remains unclear, but seems to depend on the species (cell lysis is important in *B. subtilis*, but not in other species) and furthermore likely to be based on multiple mechanisms, as various controls in fig. 2c mostly show partial effects.

I appreciate the effort taken by the authors and I think that the results are much more convincing now than in the previous version of the manuscript. Nevertheless I remain a bit skeptical, as this may not yet be the complete story.

However, the observations made here are interesting and will certainly spark follow up work. They deserve to be published.

Two minor suggestions:

1. I would suggest to tone down the title and avoid claims about redefining terms. How about "Early mechanical coupling of planktonic bacteria in dilute suspensions" or something like that.
2. I think plotting a measure of the coupling such as the coupling distance as a function of time in the same figure as a growth curve would be a nice illustration, and also nice to compare the different growth protocols.

Comments on the response to reviewer 2:

In general, reviewer 2's concern is that the extraordinary claims require extraordinary evidence. In the revision, the authors approached this issue from both sides. They have weakened or removed some strong claims as well as strengthened their evidence. The three main points criticised by the reviewer are the following (point 1 is

Collective swimming as a source of the viscoelastic behavior: The reviewer suggested as a control experiment to kill or metabolically inactivate cells and test the mechanical coupling then to separate coupling due to collective swimming from coupling by the matrix. The authors managed to do this for *B. subtilis*, but not the other species used. In *B. subtilis*, In the case of *b. subtilis*, asphyxiating them does not do much to the coupling, which argues strongly against a dominant role (if any) of collective swimming, but does not rule out a role for collective swimming entirely (or possibly in some species only?).

Rheology: Reviewer 2 and the authors have an extended discussion concerning the methods used for rheology. In the revised paper, the authors have removed the original figure 1 (which was

dominated by very high frequencies that would disrupt the matrix rather than probe its elasticity). The new experiment they include is not exactly what the reviewer suggested (classical rheology at much lower shear following the Lopez et al paper - the authors say this does not work under their conditions due to much lower cell density), but rather do microrheology with optical tweezers to measure the storage and loss modulus. This data set, even though it has large error bars shows very clearly the difference between the bacterial suspension and the pure medium and also clearly indicates viscoelastic behavior. The addition of this experiment in my opinion makes their case much stronger.

Another point here is whether there is a characteristic length of coupling. The reviewer suggests that the data might be consistent with a $1/r$ decay, thus not having a characteristic scale (and also being possibly explained by hydrodynamic coupling). The authors added a data point at larger separation that is clearly off the expected line for the $1/r$ behavior (fig. S18), but this argument rests on one data point at the limit of what can be measured. While this is evidence for the authors claim, I think this point remains a bit weak, but I consider it unlikely that more can be done.

Percolation: The percolation model to interpret the observations (which reviewer 2 considered not convincing) was removed from the paper, which I think was the right choice.

Overall, the authors have replied to the reviewer's comment in a fashion that I would consider mostly satisfying. I do not know if reviewer 2 would be satisfied, but I expect that he/she would agree that the author's argument has been improved very much, while possibly still remaining a bit skeptical about some aspects (role of collective swimming, $1/r$ behavior). However, even though mechanistic details remain open, I think some key results, the presence of a matrix and the viscoelastic behavior of the suspension have been established in this work and for sure will provide a starting point for a lot of work to follow.

Point by point reply

Reviewer #1 (Remarks to the Author):

I recognize the immense effort the authors gone through to bring the manuscript up to quality in response to my comments and recommendations. The work has elevated the quality of the manuscript vastly. I especially want to recognize the additional experiments performed with several additional species and the changes to the method. The manuscript and data are at a very high quality.

We are grateful for the reviewer comments which have significantly improved the manuscript.

Reviewer #3 (Remarks to the Author):

In the revised version of the manuscript, the authors have added a number of control experiments, specifically a comparison of different growth protocols (incl. repeated dilution) and standard microrheology measurements of viscoelasticity. They have also removed some claims that were too strong based on their experimental evidence.

*In my opinion, the paper now demonstrates clearly that the dilute bacterial suspension is viscoelastic and that mechanical coupling between cells (as defined by the optical tweezers experiment) exists. The molecular nature of this coupling remains unclear, but seems to depend on the species (cell lysis is important in *B. subtilis*, but not in other species) and furthermore likely to be based on multiple mechanisms, as various controls in fig. 2c mostly show partial effects.*

I appreciate the effort taken by the authors and I think that the results are much more convincing now than in the previous version of the manuscript. Nevertheless I remain a bit skeptical, as this may not yet be the complete story. However, the observations made here are interesting and will certainly spark follow up work. They deserve to be published.

We agree that the next step should be the elucidation of the molecular nature of the mechanical coupling which would complement the story. This may prove to be difficult, however, as various mechanisms may be responsible for the observed effect. Nevertheless, the effort is worthwhile as we will learn more about the basic fabric of the bacterial environment.

Two minor suggestions:

1. I would suggest to tone down the title and avoid claims about redefining terms. How

about “Early mechanical coupling of planktonic bacteria in dilute suspensions” or something like that.

We have toned down the title and proposed a new title “An early mechanical coupling of planktonic bacteria in dilute suspensions”.

2. I think plotting a measure of the coupling such as the coupling distance as a function of time in the same figure as a growth curve would be a nice illustration, and also nice to compare the different growth protocols.

The correlation between the two has been indeed observed. We have re-plotted a new Fig 2b where the bacterial optical density has been appended on a secondary y axis for the washed and re-suspended stationary phase cells.

Comments on the response to reviewer 2:

In general, reviewer 2’s concern is that the extraordinary claims require extraordinary evidence. In the revision, the authors approached this issue from both sides. They have weakened or removed some strong claims as well as strengthened their evidence. The three main points criticised by the reviewer are the following (point 1 is

*Collective swimming as a source of the viscoelastic behavior: The reviewer suggested as a control experiment to kill or metabolically inactivate cells and test the mechanical coupling then to separate coupling due to collective swimming from coupling by the matrix. The authors managed to do this for *B. subtilis*, but not the other species used. In *B. subtilis*, In the case of *B. subtilis*, asphyxiating them does not do much to the coupling, which argues strongly against a dominant role (if any) of collective swimming, but does not rule out a role for collective swimming entirely (or possibly in some species only?).*

Collective swimming is a well recognized phenomenon in dense bacterial suspensions, responsible for coordinated cell behavior. The collective swimming is relatively easy observed under the microscope. We did not find evidence to support the collective motion in low cell density *B. subtilis* wt suspensions. However, the reviewer is correct that we cannot rule it out completely in other bacteria. To make this statement more precise, we have slightly modified the text: “We have not observed collective swimming behavior at low cell densities in *B. subtilis* suspensions.”

Rheology: Reviewer 2 and the authors have an extended discussion concerning the methods used for rheology. In the revised paper, the authors have removed the original figure 1 (which was dominated by very high frequencies that would disrupt the matrix rather than probe its elasticity). The new experiment they include is not exactly what the reviewer suggested (classical rheology at much lower shear following the Lopez et al paper - the authors say this does not work under their conditions due to much lower cell density), but rather do microrheology with optical tweezers to measure the storage and loss modulus. This data set, even though it has large error bars shows very clearly the

difference between the bacterial suspension and the pure medium and also clearly indicates viscoelastic behavior. The addition of this experiment in my opinion makes their case much stronger.

We agree that with the new microviscoelasticity data the manuscript is much stronger. Although the data have relatively large error bars, the values for the storage modulus in the bacterial suspension are consistently higher than in the pure medium. These are, to the best of our knowledge, the first microviscoelasticity data measured in dilute bacterial suspensions which indicate the viscoelastic nature of the bacterial local environment.

Another point here is whether there is a characteristic length of coupling. The reviewer suggests that the data might be consistent with a $1/r$ decay, thus not having a characteristic scale (and also being possibly explained by hydrodynamic coupling). The authors added a data point at larger separation that is clearly off the expected line for the $1/r$ behavior (fig. S18), but this argument rests on one data point at the limit of what can be measured. While this is evidence for the authors claim, I think this point remains a bit weak, but I consider it unlikely that more can be done.

We have pushed the measurement as far as one could reliably go. Aware of the difficulty in interpretation of the hydrodynamic effect in complex solutions, we have already weakened the claim in the revised manuscript suggesting that the long-range coupling cannot be simply explained with the unscreened hydrodynamic effect.

Percolation: The percolation model to interpret the observations (which reviewer 2 considered not convincing) was removed from the paper, which I think was the right choice. Overall, the authors have replied to the reviewer's comment in a fashion that I would consider mostly satisfying. I do not know if reviewer 2 would be satisfied, but I expect that he/she would agree that the author's argument has been improved very much, while possibly still remaining a bit skeptical about some aspects (role of collective swimming, $1/r$ behavior). However, even though mechanistic details remain open, I think some key results, the presence of a matrix and the viscoelastic behavior of the suspension have been established in this work and for sure will provide a starting point for a lot of work to follow.

We do share the enthusiasm with the reviewer that there is more to come. The connections between bacteria were always there, we just did not have the right tools to investigate and observe them.